# Microtubule reorganization during female meiosis in *C. elegans*

Ina Lantzsch[1], Che-Hang Yu[2], Yu-Zen Chen[3,4], Vitaly Zimyanin[3,4], Hossein Yazdkhasti[3,4], Norbert Lindow[5], Erik Szentgyoergyi[1], Ariel M Pani[6,7], Steffen Prohaska[5], Martin Srayko[8], Sebastian Fürthauer[9]*, Stefanie Redemann[3,4,7]*

[1]Experimental Center, Faculty of Medicine Carl Gustav Carus, Technische Universität Dresden, Dresden, Germany; [2]Department of Electrical and Computer Engineering, University of California, Santa Barbara, Santa Barbara, United States; [3]Center for Membrane and Cell Physiology, University of Virginia School of Medicine, Charlottesville, United States; [4]Department of Molecular Physiology and Biological Physics, University of Virginia, School of Medicine, Charlottesville, United States; [5]Zuse Institute Berlin, Berlin, Germany; [6]Department of Biology, University of Virginia, Charlottesville, United States; [7]Department of Cell Biology, University of Virginia School of Medicine, Charlottesville, United States; [8]Department of Biological Sciences, University of Alberta, Edmonton, Canada; [9]Center for Computational Biology, Flatiron Institute, New York, United States

*For correspondence:
sfuerthauer@flatironinstitute.org
(SF);
sz5j@virginia.edu (SR)

Competing interests: The authors declare that no competing interests exist.

**Abstract** Most female meiotic spindles undergo striking morphological changes while transitioning from metaphase to anaphase. The ultra-structure of meiotic spindles, and how changes to this structure correlate with such dramatic spindle rearrangements remains largely unknown. To address this, we applied light microscopy, large-scale electron tomography and mathematical modeling of female meiotic *Caenorhabditis elegans* spindles. Combining these approaches, we find that meiotic spindles are dynamic arrays of short microtubules that turn over within seconds. The results show that the metaphase to anaphase transition correlates with an increase in microtubule numbers and a decrease in their average length. Detailed analysis of the tomographic data revealed that the microtubule length changes significantly during the metaphase-to-anaphase transition. This effect is most pronounced for microtubules located within 150 nm of the chromosome surface. To understand the mechanisms that drive this transition, we developed a mathematical model for the microtubule length distribution that considers microtubule growth, catastrophe, and severing. Using Bayesian inference to compare model predictions and data, we find that microtubule turn-over is the major driver of the spindle reorganizations. Our data suggest that in metaphase only a minor fraction of microtubules, those closest to the chromosomes, are severed. The large majority of microtubules, which are not in close contact with chromosomes, do not undergo severing. Instead, their length distribution is fully explained by growth and catastrophe. This suggests that the most prominent drivers of spindle rearrangements are changes in nucleation and catastrophe rate. In addition, we provide evidence that microtubule severing is dependent on katanin.

## Introduction

During meiosis, haploid gametes are produced from diploid progenitor cells in most sexually reproducing eukaryotes. The reduction of chromosome number in the gametes is essential for the development of the fertilized oocyte into an embryo. Therefore, the accuracy of meiosis is crucial as

errors in chromosome segregation result in aneuploidies and lead to genetic disorders, which are often lethal (*Hassold and Hunt, 2001*).

In this paper, we study the structure and dynamics of female meiotic spindles in *Caenorhabditis elegans*. Female meiotic spindles, like mitotic and male meiotic spindles, are built from microtubules (MTs), motor proteins, and microtubule-associated proteins. However, unlike other spindles in most animals, female meiotic spindles typically have no centrosomes due to centriole elimination during oogenesis. Thus, female meiotic spindles are assembled differently from those that rely on centrosome-based microtubule-organizing centers. Specifically, this is true in humans (*Holubcová et al., 2015*), mice (*Schuh and Ellenberg, 2007*), and also in nematodes (*Albertson and Thomson, 1993*). It is an open question how their structure reflects the differences in assembly.

In *C. elegans*, female meiotic spindles undergo a drastic reorganization during the metaphase-to-anaphase transition. During metaphase of meiosis I and II, microtubules form a pointed and elongated bipolar spindle, which then shortens during anaphase. In addition, the microtubule density shifts from the spindle poles to the midzone between the separating chromosomes. This midzone then extends until the end of anaphase (*Dumont et al., 2010*; *Laband et al., 2017*; *Yu et al., 2019*). The structural basis of these rearrangements and their role in meiotic chromosome segregation are not well understood. In particular, it remains unclear whether these structural rearrangements are driven by katanin-mediated severing (*Joly et al., 2016*; *Srayko et al., 2006*), transport (*Mullen and Wignall, 2017*; *Brugués et al., 2012*), or changes in MT nucleation or polymerization dynamics (*Brugués et al., 2012*; *Needleman et al., 2010*).

Here, we study the mechanisms of structural microtubule arrangement by using large-scale electron tomography (*Redemann et al., 2017*; *Redemann et al., 2018*; *Fabig et al., 2020*) to obtain the 3D ultrastructure of spindles in *C. elegans* oocyte meiosis. We find that female meiotic spindles are composed of arrays of short microtubules, 90% are shorter than half spindle length, which are highly dynamic and turnover within 10 s. During the transition from metaphase to anaphase, the number and lengths of microtubules change significantly. To understand the drivers of these changes, we developed a mathematical model for the length distributions of microtubules that includes lifetime (turnover), growth, and microtubule severing. We inferred the parameters of the model from data using Bayesian inference and Markov Chain Monte Carlo sampling. Our data suggest that the changes in microtubule number and lengths are mostly caused by changing microtubule turnover. While severing clearly occurs during metaphase, our data suggests that it affects only the small fraction of microtubules that contact the chromosomes. Here and in the following, we define that a microtubule contacts a chromosome if any point of its lattice is within 150 nm of a chromosomes surface. The rearrangements of spindle architecture towards anaphase are likely caused by changes in nucleation and not by any changes in cutting rates.

## Results

### Three-dimensional reconstruction reveals structural changes of the spindle throughout female meiosis

Previous analyses of meiotic spindle microtubules using EM tomography were based on partial, rather than full, reconstructions of the spindle. This made it impossible to measure the total length of microtubules composing the spindle since most microtubules extend beyond the tomographic volume. We obtained the spindle ultrastructure of *C. elegans* oocytes by serial section electron tomography, which allows the reconstruction of whole spindles in 3D with single-microtubule resolution (*Redemann et al., 2017*). We reconstructed two meiotic spindles in metaphase I (T0243.5 and T0243.7), as well as four spindles at early (T0243.4), mid (T0208.1), and late stages of anaphase I (T0275.10 and T0369.8; *Figure 1A*, *Figure 1—videos 1*, *2*; *Figure 3—video 1*) for meiosis I (*Table 1*). In addition, we reconstructed two metaphases (T0186.1 and T0209.7), one early anaphase (T0186.3), and one mid anaphase (T0234.6) stage of meiosis II oocytes (*Figure 1B*, *Table 2*). We had previously analyzed the modes of microtubule attachment to chromosomes (end-on vs. lateral) and discussed the resulting role of microtubules in chromosome segregation during anaphase using the meiosis I metaphase (T0243.5), early anaphase (T0243.4), mid anaphase datasets (T0208.1), (*Redemann et al., 2018*) and late anaphase (T0275.10) (*Yu et al., 2019*) of the first meiotic division. Here, we used those datasets, together with newly generated additional datasets of meiosis I

metaphase (T0243.7), and late anaphase (T0369.8), and two meiosis II metaphases (T0186.1, T0209.7), early anaphase (T0186.3), and mid anaphase (T0234.6), to quantitatively characterize the spindle structure and microtubule rearrangements during female meiosis.

For this, we compiled summary statistics that characterize spindle morphology. In *Tables 1* and *2*, we report the spindle length, the number of microtubules, the number of microtubules that are located within 150 nm from the chromosome surface, the average microtubule length, the maximum microtubule length, the total polymer length (added length of all microtubules), and the dimension

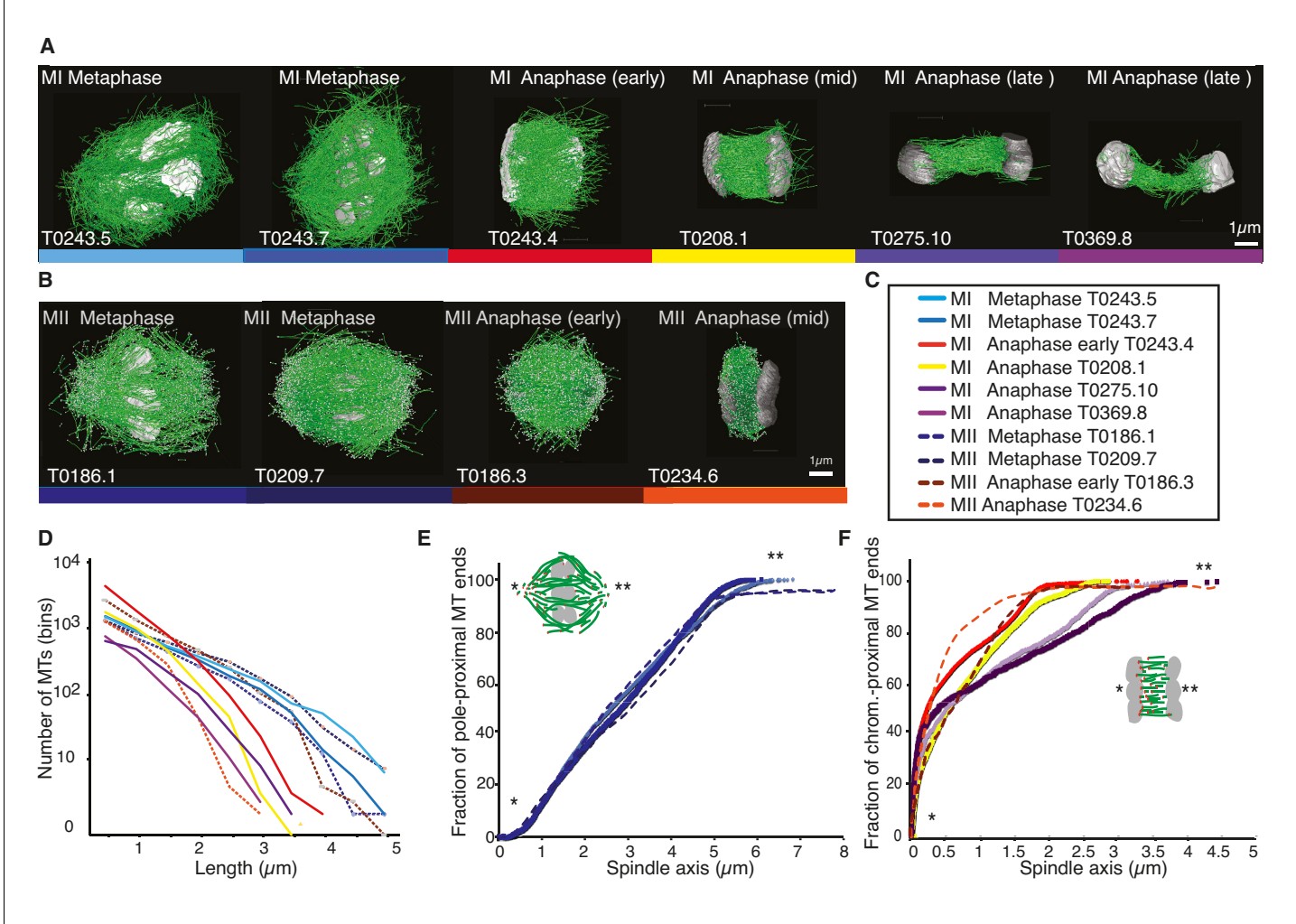

**Figure 1.** Three-dimensional organization of microtubules in meiosis I and II. (**A**) Three-dimensional models showing full reconstruction of microtubules at different stages of wild-type meiosis I. The different stages from metaphase to late anaphase are indicated. Microtubules are shown in green, chromosomes in gray. An individual color is assigned to each dataset. The anaphase datasets are oriented with the cortical side being left, cytoplasmic right. (**B**) Like (**A**) but three-dimensional models showing full reconstruction of microtubules at different stages of wild-type meiosis II. (**C**) Legend for the different datasets plotted in (**D–F**). (**D**) Length distribution of microtubules composing the different spindles. Bin size 250 nm. (**E**) Cumulative distance function of the pole-proximal microtubule endpoints in metaphase of meiosis I and II. The position of the spindle poles in the schematic drawing and the datasets is indicated (stars). (**F**) Cumulative distance function of the chromosome-proximal microtubule endpoints in anaphase. The position of the poles is indicated (stars).

The online version of this article includes the following video(s) for figure 1:

**Figure 1—video 1.** Visualization of spindle ultrastructure in meiosis I at mid anaphase.

https://elifesciences.org/articles/58903#fig1video1

**Figure 1—video 2.** Visualization of spindle ultrastructure in meiosis I at late anaphase.

https://elifesciences.org/articles/58903#fig1video2

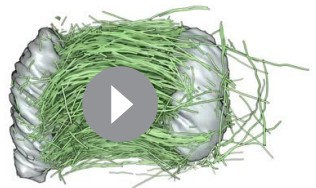

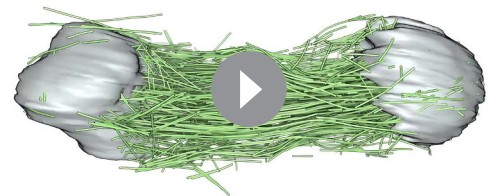

**Video 1.** FRAP during metaphase in Meiosis I in control embryo. Movie of a FRAP experiment during metaphase in meiosis I of *C. elegans* expressing GFP tubulin and mCherry Histone. A small stripe is bleached next to the Chromosomes (right side) and the motion and recovery of the stripe over time are analyzed. Framerate 1fps.

https://elifesciences.org/articles/58903#video1

**Video 2.** FRAP during metaphase in Meiosis I in *mei-2 (ct98)* embryo. Movie of a FRAP experiment during metaphase in meiosis I of *C. elegans mei-2(ct98)* expressing GFP tubulin and histone cherry. Only the tubulin channel is shown. A small stripe is bleached in the center of the spindle and recovery of the stripe over time is analyzed. Framerate five fps.

https://elifesciences.org/articles/58903#video2

of the tomogram and spindle dimensions for each stage of meiosis I (*Table 1*) and II (*Table 2*). We also indicated new and previously analyzed datasets in this table.

## Female meiotic spindles are arrays of short microtubules

We first sought to characterize the overall spindle structure. Female meiotic spindles could be constructed mainly from long microtubules, reaching from near the spindle pole to chromosomes (as in *C. elegans* mitotic spindles; *Redemann et al., 2017*), or from short ones distributed throughout the spindle volume, as suggested by *Srayko et al., 2006*, and, for instance, meiotic *Xenopus* spindles (*Brugués et al., 2012*; *Burbank et al., 2006*; *Yang et al., 2007*).

Our data show that throughout meiosis I and meiosis II, roughly 40–50% of microtubules are shorter than 500 nm (*Figure 1D*). In contrast, the fraction of microtubules that are at least half spindle length or longer is only 10%. Throughout meiosis, the mean length of microtubules is approximately 20% of the spindle length (*Tables 1* and *2*).

Furthermore, we determined the position of the pole-proximal ends of microtubules along the spindle axis. In metaphase, these putative microtubule minus-ends were found throughout the spindle (*Figure 1E*), with no marked preference for the poles. In anaphase, about 60–70% of microtubules are found within a distance of 150 nm from the chromosomes, and the others are distributed throughout the spindle (*Figure 1F*).

Together, these findings directly confirm that the acentrosomal female meiotic *C. elegans* spindle is an array of short microtubules, more similar to meiotic *Xenopus* spindles than to mitotic spindles in *C. elegans*, as previously inferred by others based on light microscopy (*Brugués et al., 2012*; *Burbank et al., 2006*; *Yang et al., 2007*). We conclude that the distance between poles in metaphase (5 µm) and chromosomes in anaphase (3–4 µm) is bridged by an array of short overlapping microtubules rather than long ones spanning the entire distance. This is an overall interesting finding as *C. elegans* meiotic spindles are rather short in comparison to *Xenopus* meiotic spindles and could easily generate microtubules long enough to bridge the entire spindle length, which is much harder to achieve in a 60 µm *Xenopus* spindle.

## Meiotic spindles are composed of short-lived, fast-moving microtubules

We next sought to understand the role of microtubule movements and dynamics in the structural changes observed during the transition from metaphase to anaphase. Since electron tomography generates static snapshots, we investigated this question by light microscopy. Using fluorescence recovery after photobleaching (FRAP), we first measured microtubule turnover and motion in metaphase I spindles. By photobleaching a small stripe near the center as well as close to the poles of metaphase spindles (*Figure 2A*, arrows, *Video 1*), we observed a half-time of microtubule recovery in metaphase of 4.9 s ± 3.4 s (n = 6) in the spindle center and 4.1 s ± 2.7 s (n = 6) at the spindle poles

**Table 1.** Summary of quantifications of tomographic data for meiosis I.
The spindle length, the total number of microtubules, the number of microtubules within 150 nm from the chromosome surface, the average microtubule length, the maximum length of microtubules, the total length of all microtubules (net polymer length) in the spindle, and the dimension of the tomogram and spindle dimensions are shown for each tomographic dataset. The * labels newly generated datasets. MTs: microtubules.

| Meiosis I | Metaphase T0243.5 | Metaphase T0243.7* | Early anaphase T0243.4 | Mid anaphase T0208.1 | Late anaphase T0275.10* | Late anaphase T0369.9* |
|---|---|---|---|---|---|---|
| Spindle length (µm) | 5 | 5.4 | 3.1 | 2.7 | 4.5 | 5.1 |
| Number of MTs | 3662 | 3812 | 7011 | 3317 | 1511 | 1306 |
| MT within 150 nm | 500 | 920 | 5361 | 2334 | 875 | 699 |
| Average MT length (µm) | 0.91 ± 0.08 | 1.04 ± 0.93 | 0.58 ± 0.52 | 0.62 ± 0.49 | 0.74 ± 0.54 | 0.55 ± 0.45 |
| Max. MT length (µm) | 5 | 5.4 | 5 | 3.2 | 4.2 | 2.9 |
| Net polymer length (µm) | 3338 | 3961 | 4045 | 2061 | 1111 | 724 |
| Tomogram dimensions (µm) | 9.1 × 9.7 × 3.8 | 9.1 × 8 × 2.1 | 5 × 5.9 × 2.8 | 5.5 × 6.6 × 4.1 | 7.5 × 5.6 × 2.6 | 6.9 × 5.5 × 2.6 |
| Spindle dimensions (µm) | 5 × 8 × 3.9 | 5.4 × 5.6 × 2.1 | 3.1 × 4.3 × 2.6 | 2.7 × 3 × 2.5 | 4.5 × 1.2 × 1 | 5.1 × 1.4 × 0.8 |

(*Figure 2B*). In addition, we observed that the photobleached stripes close to the spindle poles showed a rapid poleward movement with a rate of 8.5 ± 2.2 µm/min (n = 6, *Figure 2C*).

We conclude that microtubules in *C. elegans* meiosis turn over rapidly and show substantial poleward motion. This is notably different from the first mitotic division in the early *C. elegans* embryos, where microtubule motion was not detected in similar experiments (*Labbé et al., 2004*; *Redemann et al., 2017*). Thus, in female meiosis, both nucleation/depolymerization and microtubule motion might be involved in shaping the spindle structure. Similar microtubule dynamics have been described in *Xenopus* meiotic spindles using fluorescent speckle microscopy (*Burbank et al., 2006*; *Yang et al., 2007*; *Burbank et al., 2006*; *Yang et al., 2007*).

## Spindle rearrangements during meiosis correlate with substantial changes in microtubule number

To better understand the role of microtubule polymerization dynamics, we next asked whether the number of spindle microtubules changed during spindle rearrangements. From our reconstructions, we detected 3662 and 3812 microtubules in metaphase of meiosis I (3013 and 3808 in meiosis II), 7011 microtubules in early anaphase of meiosis I (5572 in meiosis II), 3317 in mid anaphase of

**Table 2.** Summary of quantifications of tomographic data for meiosis II.
The spindle length, the total number of microtubules, the number of microtubules within 150 nm from the chromosome surface, the average microtubule length, the maximum length of microtubules, the total length of all microtubules (net polymer length) in the spindle, and the dimension of the tomogram and spindle dimensions are shown for each tomographic dataset. The * labels newly generated datasets. MTs: microtubules.

| Meiosis II | Metaphase T0186.1* | Metaphase T0209.7* | Early anaphase T0186.3* | Mid anaphase T0234.6* |
|---|---|---|---|---|
| Spindle length (µm) | 5.9 | 5.8 | 4.0 | 2.5 |
| Number of MTs | 3013 | 3808 | 5572 | 2246 |
| MT within 150 nm | 484 | 887 | 3298 | 1359 |
| Average MT length (µm) | 0.88 ± 0.78 | 1.11 ± 0.92 | 0.90 ± 0.79 | 0.57 ± 0.49 |
| Max. MT length (µm) | 4.7 | 5.8 | 4.6 | 2.8 |
| Net polymer length (µm) | 2693 | 4237 | 4512 | 236 |
| Tomogram dimensions (µm) | 8.6 × 9.8 × 3.3 | 11 × 5.5 × 2.8 | 5.6 × 6.9 × 2.7 | 9 × 6.2 × 3.3 |
| Spindle dimensions | 5.9 × 5.8 × 3.8 | 5.8 × 4.1 × 3.1 | 4 × 4.1 × 3.3 | 2.5 × 3.7 × 2.8 |

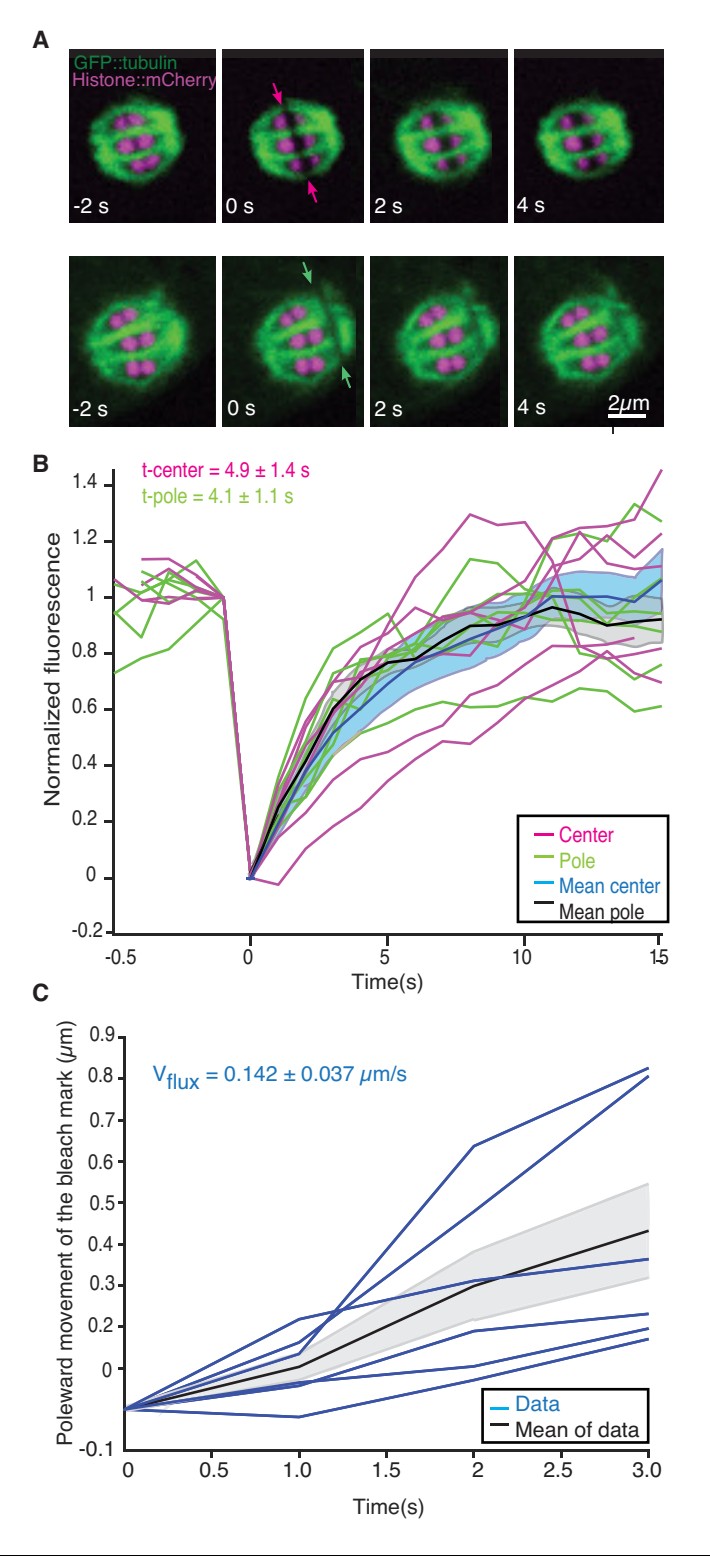

**Figure 2.** Microtubule dynamics during metaphase of meiosis I. (**A**) Light microscope images of spindles in meiosis I prior to and after photobleaching at the spindle center (top row, red arrows) and close to the spindle pole (bottom row, green arrows). The bleaching of the sample (t = 0) and the frame rate is indicated. (**B**) Plot of the recovery of the bleach mark over time at the center (magenta, mean values given in blue) and pole (green, mean values given in black) for different datasets. The recovery times for the datasets are shown in the plot. (**C**)

*Figure 2 continued on next page*

*Figure 2 continued*

Plot showing the poleward motion of the bleach mark at the spindle pole over time for different datasets (blue). The mean is indicated in black.

---

meiosis I (2246 in meiosis II), and 1511 and 1306 microtubules in late anaphase of meiosis I (*Tables 1* and *2*). This shows that the number of microtubules almost doubles between metaphase and early anaphase. To get a more spatially resolved picture, we also measured the number of microtubules along the spindle axis in our datasets. This measurement showed that the number of microtubules increases by more than twofold in the center of the spindle from metaphase to early anaphase, while the number of microtubules at the spindle poles only slightly increases from metaphase to early anaphase and then decreases throughout anaphase (*Figure 3A*). During anaphase, the number of microtubules decreased on the poles and almost halved in the spindle center (*Figure 3A, B*). Thus, we conclude that changing the number of microtubules is correlated with the structural spindle rearrangements that occur during meiosis.

## Microtubule rearrangements during meiosis are driven by changes in microtubule dynamics, not nucleation rates alone

We next sought to gain insight into the mechanisms that drive the observed changes in microtubule numbers. We first asked whether changes in nucleation rate alone could explain the data. If this were the case, microtubule growth, shrinkage, and severing rates would remain the same, and the average length of the microtubules would remain unchanged throughout meiosis. However, the total polymer amount, the length of all microtubules added together, would change in step with the microtubule number.

Interestingly, we found the opposite result: the total polymer length of microtubules remained almost constant from metaphase to anaphase even though the number of microtubules increases twofold (*Tables 1* and *2*). Furthermore, during the same time, the average length of microtubules went down by a factor of 2 (*Figure 1D*, *Tables 1* and *2*). Only in late anaphase, as the spindle continued to elongate, did the number of microtubules and the total polymer length start to decrease. Therefore, the observed changes in spindle morphology were not caused by changes in nucleation rates alone, but also by changes in microtubule disassembly. In principle, this could be caused by local or global changes in microtubule severing, changes in microtubule growth dynamics, or a combination thereof. In the following, we will try to disentangle these possibilities.

## Microtubule dynamics during spindle rearrangement are globally regulated

We next investigated whether changes in microtubule length were uniform along the spindle axis by quantifying the average microtubule length along the spindle axis. This analysis revealed that, during the transition from metaphase to early anaphase, the average length of microtubules decreased everywhere in the spindle (*Figure 3C, D*). However, the magnitude of the decrease in length varies as a function of position. The average length of microtubules in the middle of the spindle decreased nearly twice as much as the average length of microtubules near the spindle poles (*Figure 3C, D*). The analysis of average microtubule length along the axis also showed that in all spindles, in metaphase and anaphase, the average length of microtubules is shorter near the spindle poles (*Figure 3C, D*), which in anaphase is located at the inner surface of the chromosomes.

We more closely mapped the position of microtubules according to their length (*Figure 4A–G*) by plotting the number of microtubules that are 500 nm and shorter as well as the number of microtubules between 0.5–1 μm and 1–1.5 μm along the spindle axis (*Figure 4H–N*). This analysis showed that short microtubules, 500 nm and below, are localized closer to the spindle poles, while longer microtubules, between 0.51 and 1.5 μm, are localized to the spindle center. We also found that the number of short microtubules increased from approximately 35% of all microtubules in metaphase to 50% of all microtubules in anaphase (*Figure 4A–G*).

Throughout anaphase, we noticed that there were more short microtubules in the spindle half closer to the cell cortex (*Figures 3* and *4*). This difference was most apparent in early and mid anaphase, where the average microtubule length in the cortical spindle half was 1.00 μm ± 0.6 (p=3.6

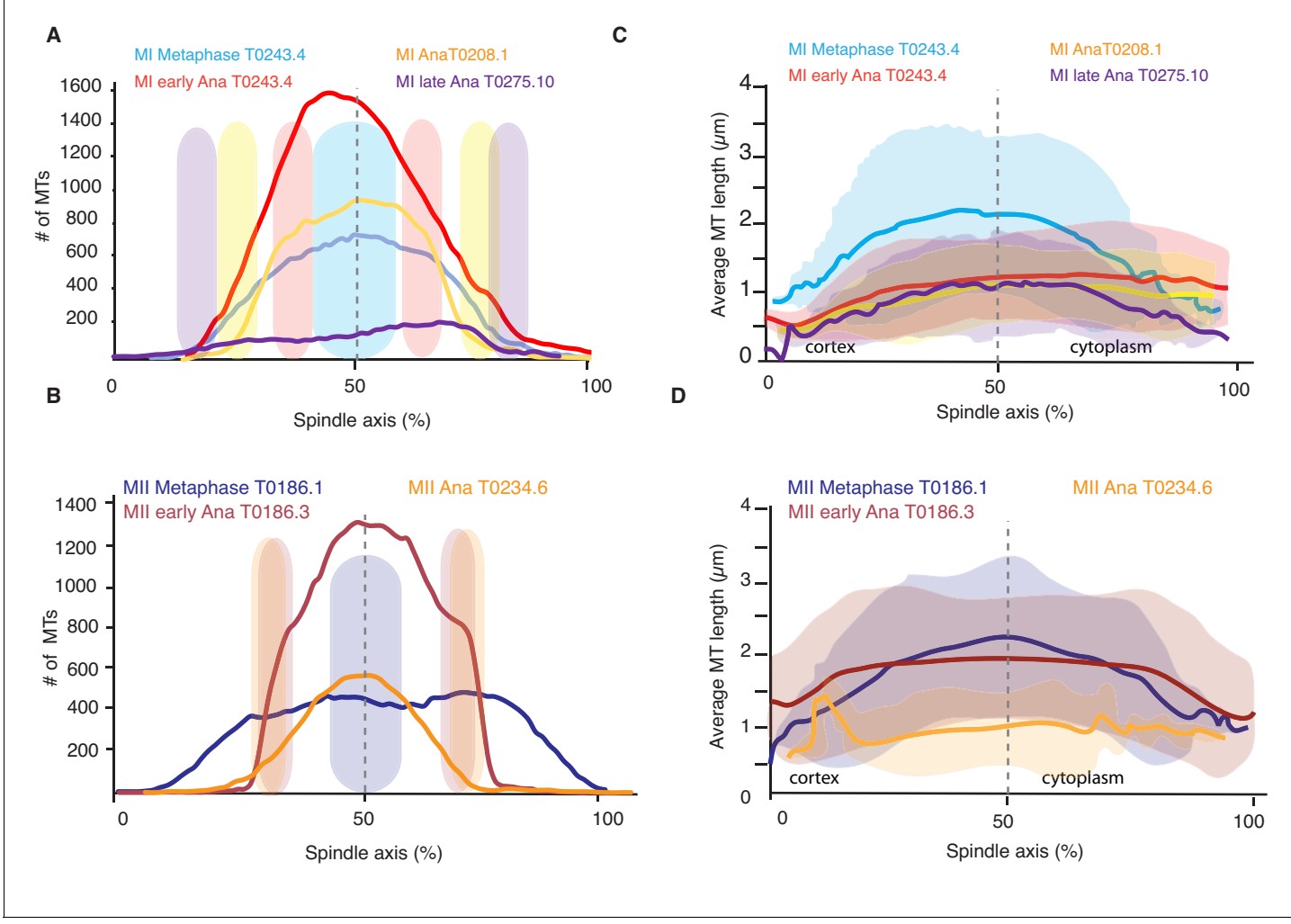

**Figure 3.** Analysis of microtubule number and length along the spindle axis. (A) Plot of the number of microtubules at different positions (100 nm steps) along the spindle axis for four different datasets in metaphase, early anaphase, and mid and late anaphase in meiosis I. The approximate position of chromosomes for each dataset is indicated by the colored ovals. The datasets are oriented with '0' being at the cortical side. Datasets are aligned with the center of the spindle located at 50%. (B) Same plot as in (A) but for meiosis II. (C) Plot of the average microtubule length at different positions (100 nm steps) along the spindle axis for four different datasets in metaphase, early anaphase, and mid and late anaphase. Shaded color indicates the standard deviation, datasets are oriented with '0' being at the cortical side. Datasets are aligned with the center of the spindle located at 50%. (D) Same plot as in (C) but for meiosis II.

The online version of this article includes the following video for figure 3:

**Figure 3—video 1.** Visualization of spindle ultrastructure in meiosis I at late anaphase.

https://elifesciences.org/articles/58903#fig3video1

E-6) and 1.17 μm ± 0.8 in the cytoplasmic half, and in early anaphase with an average length of 0.91 ± 0.53 (p=0.002) in the cortical half and 1.04 μm ± 0.6 in the cytoplasmic half. We do not know at this point what causes this asymmetry.

Together, our data suggest that microtubule dynamics change throughout the spindle during spindle rearrangements. While the magnitude of this effect is spatially varying, throughout the spindle nucleation and turnover rates increase.

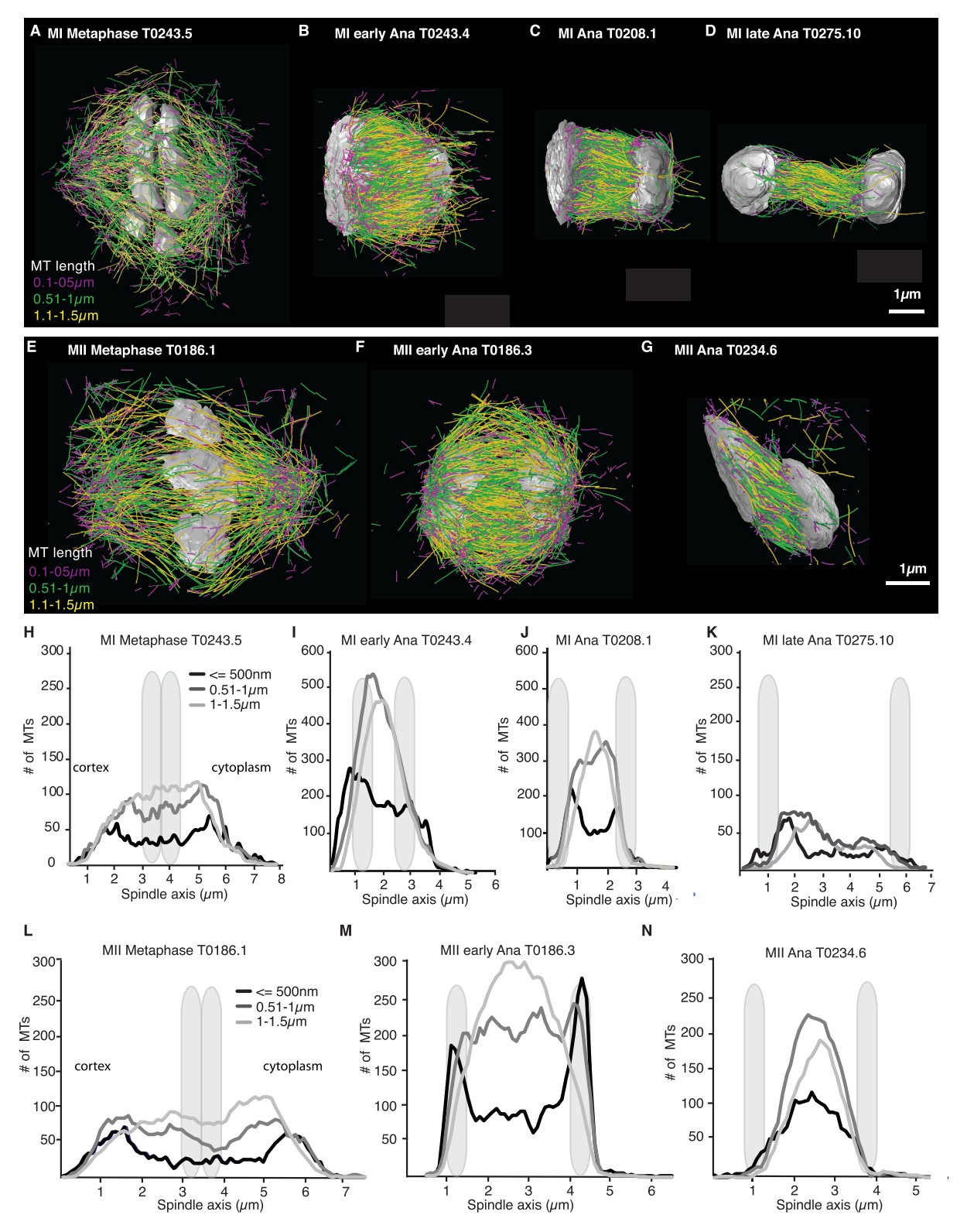

**Figure 4.** Analysis of microtubule number according to their length along the spindle axis. (A–G) Tomographic reconstructions of meiotic spindles in meiosis I and II showing microtubules of different length. Microtubules with length ≤ 500 nm are magenta, microtubules between 0.51 and 1 μm are green, and microtubules between 1.1 and 1.5 μm are yellow. Chromosomes are gray, spindles are oriented with the cortical site to the left. Scale bar 1 μm. (H–N) Plots showing the number of microtubules according to their length along the spindle axis for the datasets shown in (A–G). Dark line shows

*Figure 4 continued on next page*

*Figure 4 continued*

the number of microtubules ≤ 500 nm, medium gray line microtubules between 0.5 and 1 µm, and light gray lines microtubules between 1 and 1.5 µm. The approximate position of chromosomes is indicated by gray outlines.

## Bayesian inference on a mathematical model for microtubule dynamics reveals the relative importance of microtubule severing and growth dynamics for the global spindle dynamics

We next sought to better understand the processes that drive the observed microtubule length changes during spindle rearrangements. For this, we constructed a mathematical model that solves for the expected microtubule length distribution given their growth velocity $v_g$ their rate of undergoing catastrophe $r$, and the rate at which microtubules are cut per unit time and length $\kappa$. The stability of microtubule plus ends created by cutting is encoded in the parameter $\alpha$, which interpolates between the two extreme cases, where newly created plus ends immediately depolymerize ($\alpha = 0$) and where newly created plus ends are indistinguishable from preexisting ones ($\alpha = 1$; see *Figure 5*). The mathematical details of this model are given in Material and methods. In spirit, it is very similar to the model proposed in *Kuo et al., 2020*.

We used this mathematical model to infer the relative importance of microtubule cutting and changes in microtubule nucleation. Using Markov Chain Monte Carlo sampling (*Foreman-Mackey et al., 2013*; *MacKay, 2003*), we infer that the most likely values for the dimensionless ratios $\bar{r} = r/v_g\ell$ and $\bar{\kappa} = \kappa/v_g\ell^2$ predict the location of the average microtubule lengths $\ell$, given the experimentally determined microtubule length distribution (see Materials and methods). Importantly this inference scheme only uses the experimentally measured microtubule length distribution as input and allows us to quantify the relative importance of cutting to turnover $\bar{r}/\bar{\kappa}$ independently of direct measurements of turnover rates and microtubule growth velocities, which are hard to do in these very small spindles. We note that additional experiments like FRAP allow us to give absolute time units since it directly measures the turnover rate. The results of the inference scheme are however independent of the FRAP measurements. For details, we refer the reader to Materials and methods.

Using this approach, we first sought to infer the relative importance of turnover to cutting throughout the spindle. In *Figure 6*, we show the inferred posterior probabilities of model parameters based on the data (*Figure 6A–D*, *Figure 6—figure supplement 1A–D*). These plots give the

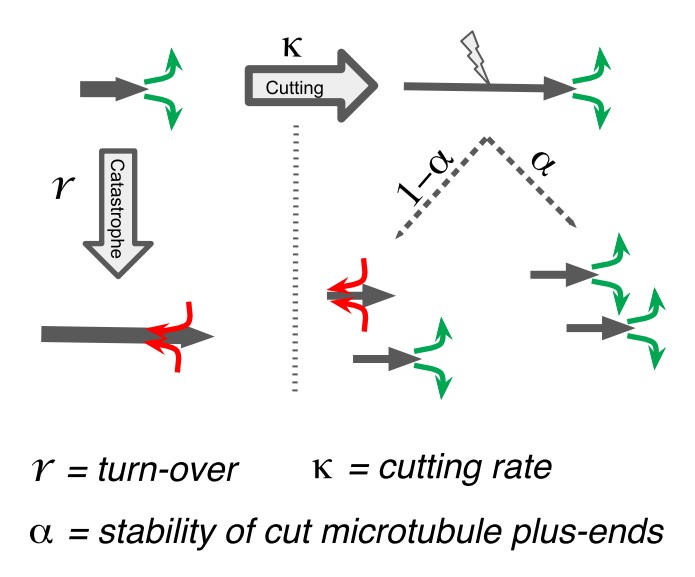

**Figure 5.** Processes determining microtubule (MT) length distributions. Our model considers MT turnover with a rate $r$ and cutting with a rate $\kappa$. $\alpha$ is the stability of cutting-generated MT plus ends.

most likely values of the model parameters given the data, 95% confidence intervals for each of the parameters, and 2D error plots for each pair of parameters under the assumption that the third is at its most likely value. We also compared the predictions of the model parameterized by the expectation values of all parameters to our data (*Figure 6E–H*, *Figure 6—figure supplement 1E–H*).

We asked whether the transition of spindle structure from metaphase to early anaphase can be explained by microtubule turnover, which here means changes in growth rate and catastrophe, or microtubule cutting. We report our results in terms of the non-dimensionalized turnover rate $\bar{r} = r/v_g\ell$ and the non-dimensionalized cutting rate $\bar{\kappa} = \kappa/v_g\ell^2$, where $\ell$ is the average microtubule length. Most importantly, the ratio of these two quantities $\bar{r}/\bar{\kappa}$ gives the length (in units of average MT length) a microtubule has to reach for it to be more likely to get severed than to undergo catastrophe. This quantifies the relative importance of turnover to cutting; for example, if $\bar{r}/\bar{\kappa}$ is much smaller than 1, then this indicates a length distribution that is dominated by cutting; if $\bar{r}/\bar{\kappa}$ is much larger than 1, it indicates a length distribution that is dominated by catastrophe and turnover. In metaphase (*Figure 6A, B, E, F*, *Figure 6—figure supplement 1A, B, E, F*) and early anaphase (*Figure 6C, G*, *Figure 6—figure supplement 1C, G*), we find that the non-dimensionalized turnover rate $\bar{r} = r/v_g\ell$ is equal to 1.0, while the non-dimensionalized cutting rate $\bar{\kappa} = \kappa/v_g\ell^2$ is about 0.1. Here, $\ell$ is the average microtubule length. These numbers imply that the length at which a microtubule is more likely to be cut than to undergo catastrophe is $\bar{r}/\bar{\kappa} \simeq 10$, that is, cutting is a relatively rare event. In fact, there are no microtubules in metaphase that are longer than 10 times the average microtubule length (i.e., 10 µm) and only one microtubule was larger than *10 times the average length* (i.e., 5 µm) in early anaphase. This also implies that the change in the microtubule length observed from metaphase ($\bar{r}/\bar{\kappa} \simeq 10 = 1$ µm) to early anaphase ($\ell = 0.5$ µm) is a consequence of either the microtubule turnover rate or changes in microtubule growth velocity. For error estimates

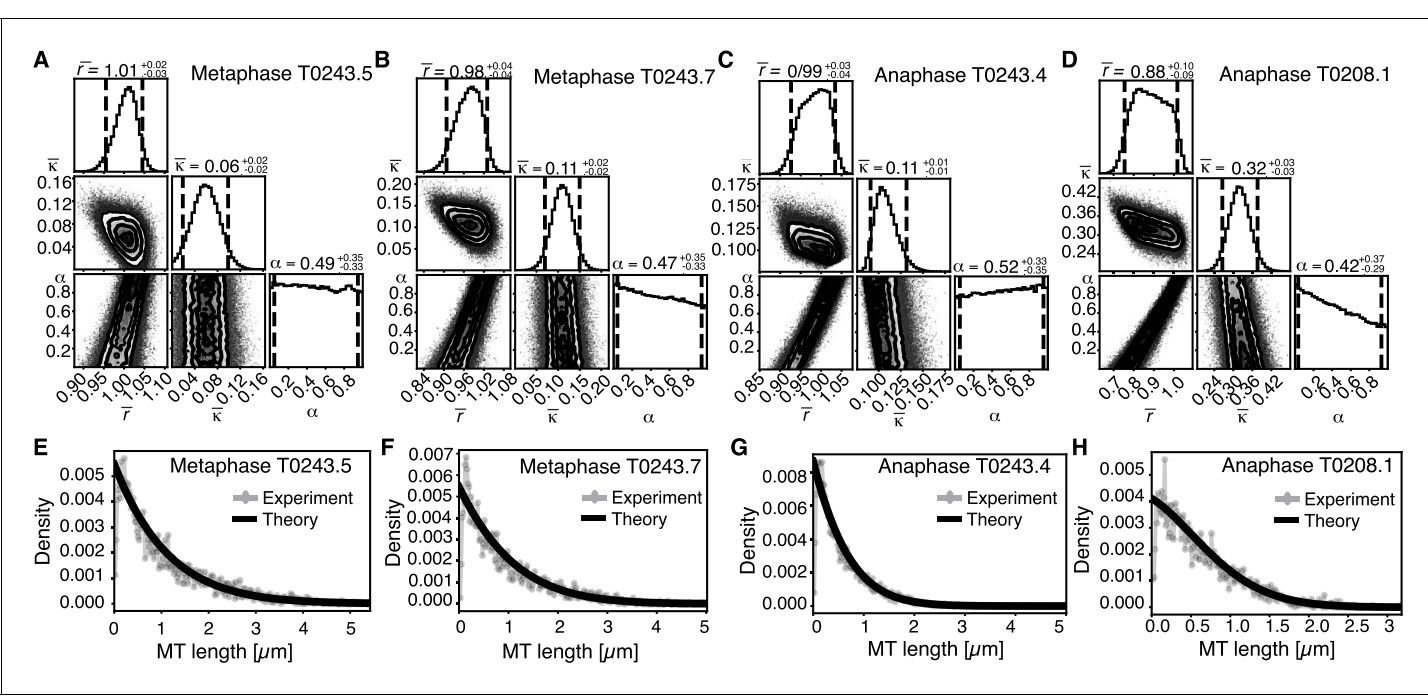

**Figure 6.** Inference of model parameters in metaphase and anaphase spindles of meiosis I. (A) Likelihood distribution of model parameters determined by Markov Chain Monte Carlo (MCMC) for metaphase T0243.5. The top boxes show the totally marginalized distribution of parameters, with dashed lines delimiting the 95% confidence interval. Surface plots show cuts through the likelihood distributions, marginalized onto 2D subspaces. Lines are contour lines, dots indicate MCMC samples. (B) Same as in (A) but for the metaphase T0243.7 dataset. (C) Same as in (A) but for the early anaphase T0243.4 dataset. (D) Same as in (A) but for the mid anaphase T0208.1 dataset. (E–H) Comparison of experimentally determined length distribution of microtubules (dots) to the prediction of the highest likelihood model (solid line) for metaphase T0243.5 (E), metaphase T0243.7 (F), early anaphase T0243.4 (G), and mid anaphase T0208.1 (H). Plots for meiosis II can be found in *Figure 6—figure supplement 1*.

The online version of this article includes the following figure supplement(s) for figure 6:

**Figure supplement 1.** Inference of model parameters in metaphase and anaphase spindles of meiosis II.

and detailed likelihood plots of the inferred parameters, see *Figure 6* and *Figure 6—figure supplement 1*, which give 95% confidence intervals and likelihood surfaces for all parameters. Note also that given that cutting occurs very rarely the inference is not informative on the parameter α, which describes the stability of newly cut microtubules.

We next use our estimate for the ratio $\bar{r} = r/v_g\ell$ to infer the microtubule growth velocity in meiotic metaphase spindles. For this, we took the measured FRAP half-life in metaphase (see *Figure 2B*) of about 5 s as an estimate for microtubule turnover time $1/r$ and determine the average microtubule length of about 1 µm (*Tables 1* and *2*). Using these values, we estimated that the microtubule growth velocity (vg) during metaphase was close to 12 µm/min. Interestingly, this agrees with growth velocity estimates based on microtubule flux, and it is similar to values reported for microtubule growth rates in the dense regions of mitotic *C. elegans* spindles, reaching from 16 to 20 µm/min (*Redemann et al., 2017*). Notably, this value is slower than the growth rates reported for astral microtubules (44 µm/min) growing away from centrosomes reported by *Redemann et al., 2017* and *Srayko et al., 2005*. This suggests that microtubule growth velocities are under spatial and temporal control within the cytoplasm, as was previously described by *Geisterfer et al., 2019* and *Walczak et al., 2016*.

The drastic change in the average microtubule length from metaphase to anaphase could be due to a decrease in microtubule growth velocity to half its metaphase value (6 µm/min). Alternatively, it could also be caused by an increase in microtubule turnover rate. Given the large relative errors of the FRAP measurements in the very small meiotic spindles, we cannot definitely answer whether microtubule growth speed, turnover rate, or both are under biochemical control at this stage. In toto, our data argue that the transition from metaphase to early anaphase is best explained by modulating microtubule growth and polymerization dynamics, and not by increased microtubule cutting. Note that a role for katanin in amplification of microtubule mass and number has been suggested in vitro, where severing activity can result in the incorporation of GTP tubulin into the microtubule lattice, thus increasing the rescue frequency and stability of newly emerging microtubule ends (*Vemu et al., 2018*).

We finally asked the same question for the transition from early to mid anaphase. We find that later in anaphase the relative importance of cutting increases with $\bar{\kappa} = 0.3$ and $\bar{r} = 0.9$ (see *Figure 6D, H*). This means that, at this later point, the length at which a microtubule is more likely to be cut than to undergo catastrophe decreases to 3 $\ell$, which would be approximately 3% of the microtubule population.

Thus, our data suggest that the initial steps of the transition from metaphase to anaphase are due to changes in microtubule turnover rate and growth and not mediated by katanin-dependent severing. This is surprising since earlier work clearly demonstrated the importance of severing during spindle assembly (*Srayko et al., 2006*; *McNally and McNally, 2011*; *McNally et al., 2014*; *Connolly et al., 2014*; *Joly et al., 2016*). Estimates of the number of katanin-mediated cutting events by counting lateral defects in partial EM reconstructions of meiotic spindles at earlier stages had found a large number of cutting events in wildtype spindles (*Srayko et al., 2006*). To test the predictions of our inference scheme, we decided to look for similar cutting sites in our datasets.

Analyzing the frequency of lateral defects in our tomographic data indicated a very low abundance (1.1% of microtubules in metaphase #2, 2.5% in early anaphase, and 1.8% in mid anaphase). We could also not detect an increased occurrence of lateral defects in longer microtubules or at distinct positions within the spindle. This is notably fewer events than what was reported for earlier spindles. We conclude that the initial steps of the transition from metaphase to anaphase are due to changes in microtubule turnover rate and growth and not mediated by katanin-dependent severing.

## Cutting selectively occurs for microtubules in close chromosome contact

Our finding that global rearrangements of the spindle structure are mainly caused by changes in nucleation and turnover dynamics made us wonder about the role of katanin-mediated cutting in these spindles. To investigate this, we separated out microtubules in close contact to chromosomes (defined as being closer than 150 nm) (*Figure 7A, B*) and analyzed them separately using our inference scheme. The results are shown in *Figure 8* and *Figure 8—figure supplement 1*. In metaphase spindles, the fraction of microtubules in close contact to chromosomes is small. We found 13, 25, 16, and 23% for our four metaphase spindle datasets, respectively. However, in contrast to the global

population, they showed strong evidence of cutting and $\bar{\kappa} > 1$ in all four examples. Furthermore, the length distribution of these microtubules, as depicted in *Figure 7C*, *Figure 8E,,F*, and *Figure 8—figure supplement 1E, F*, is clearly not exponential in metaphase, which is also indicative of cutting.

In contrast, in anaphase, the majority of microtubules (about 70%) are in close contact to chromosomes. Despite this, even microtubules in close contact with chromosomes showed a nearly exponential length distribution (*Figure 7D*, *Figure 8G, H*, and *Figure 8—figure supplement 1G, H*), and

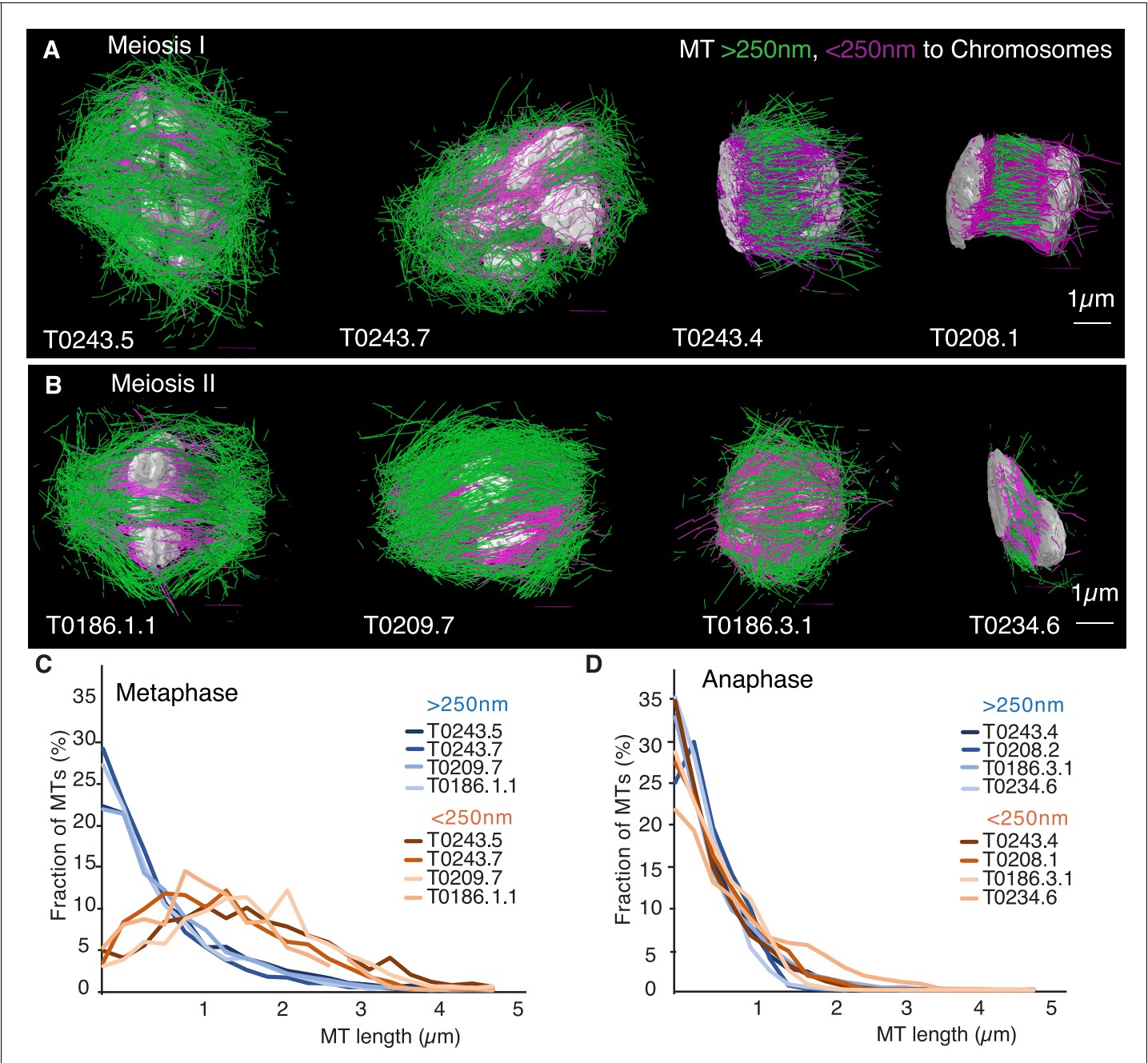

**Figure 7.** Analysis of microtubule length distributions within 150 nm of the chromosomes. (A) Tomographic reconstructions of meiotic spindles in meiosis I from metaphase to anaphase. The reconstructions show microtubules located with and end or lattice point within 150 nm from the chromosome surface in magenta and microtubules located further away in green. Chromosomes are gray. Scale bar 1 μm. (B) Same as (A) but for spindles in meiosis II. (C) Plot of the length distribution of microtubules in respect to their distance from the chromosomes for metaphase spindles in meiosis I and II. Red lines show microtubules located within 150 nm from the chromosomes, blue lines show microtubules further away. Bin size 250 nm. (D) Similar to the plot in (C) but for anaphase spindles in meiosis I and II.

our inference scheme yields the same results for microtubules near chromosomes and the global microtubule population. The plots for microtubules that were located further than 150 nm are comparable to the plots for all microtubules (*Figure 6*) and are shown in *Figure 8—figure supplement 2*.

Together, these data suggest a role for cutting in maintaining the metaphase spindle. In particular, we hypothesize that katanin selectively cuts microtubules that are in close contact with chromosomes. This mechanism seems to be suppressed in late anaphase.

## Katanin depletion reduces the fraction turnover but not the turnover rate of microtubules

We next sought to investigate the role of katanin on microtubule turnover in the metaphase spindle. As null mutants of the *C. elegans* katanin homologs MEI-1 and MEI-2 do not assemble bipolar spindles (*Mains et al., 1990*), we decided to work with the FM13 *mei-2(ct98)* strain, which has an overall lower expression of MEI-2 (*Srayko et al., 2000*; *O'Rourke et al., 2011*) and reduced microtubule-severing activity (*McNally et al., 2006*; *McNally et al., 2014*). We used FRAP experiments to measure the effect of katanin mutation on microtubule turnover using the FM13 *mei-2(ct98)* strain (*Figure 9A, B*, *Video 2*). We found that the characteristic time scales for recovery were not

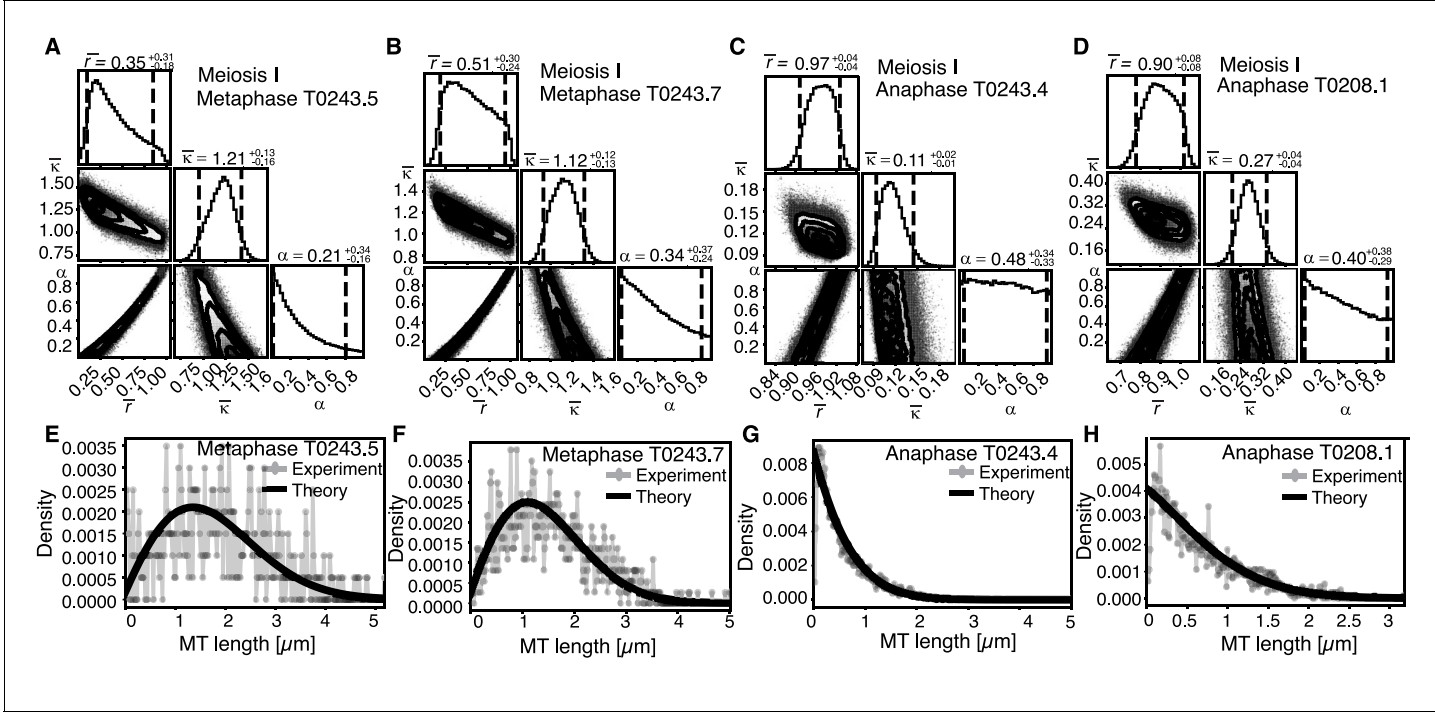

**Figure 8.** Selective inference model parameters in metaphase and anaphase spindles in meiosis I for microtubules located within 150 nm of the chromosomes. (A) Likelihood distribution of model parameters determined by Markov Chain Monte Carlo (MCMC) for microtubules within 150 nm distance from the chromosomes in metaphase T0243.5 meiosis I. The top boxes show the totally marginalized distribution of parameters, with dashed lines delimiting the 95% confidence interval. Surface plots show cuts through the likelihood distributions, marginalized onto 2D subspaces. Lines are contour lines, dots indicate MCMC samples. (B) Same as in (A) but for meiosis I metaphase T0243.7 dataset. (C) Same as in (A) but for meiosis I T0243.3 early anaphase dataset. (D) Same as in (A) but for the meiosis I mid anaphase T0208.1 dataset (E–H) Comparison of experimentally determined length distributions of microtubules located within 150 nm from the chromosome surface (dots) to the prediction of the highest likelihood model (solid line) for meiosis I metaphase T0243.5 (E), meiosis I metaphase T0243.7 (F), meiosis I early anaphase T0234.4 (G), and meiosis I mid anaphase T0208.1 (H). Plots for meiosis II can be found in *Figure 8—figure supplement 1*, and plots for microtubules further away than 150 nm can be found in *Figure 8—figure supplement 2*.

The online version of this article includes the following figure supplement(s) for figure 8:

**Figure supplement 1.** Selective inference model parameters in meiosis II metaphase and anaphase spindles for microtubules located within 150 nm of the chromosomes.

**Figure supplement 2.** Selective inference model parameters in metaphase and anaphase spindles for microtubules located further than 150 nm from the chromosomes in meiosis I and II.

significantly affected (4.5 s ± 1.8 s [n = 7] in control and 2.8 s ± 3.4 s [n = 11] in mei-2(ct98), p=0.2434). However, strikingly we observed a significant decrease in the fraction of fluorescence recovery of microtubules in the mei-2(ct98) embryos (*Figure 9C*). While in wildtype embryos 87% ± 10% of the fluorescent tubulin signal recovers, this is significantly reduced to 52% ± 26% (p=0.0039) in the katanin mutant. This suggests the presence of a population of stabilized microtubules in katanin-depleted spindles. In line with other authors (*Brinkley and Cartwright, 1975*; *Rieder, 1981*; *Bakhoum et al., 2009*), we thus speculate that microtubules in close contact to chromosomes get stabilized by interacting with the chromosomes. Katanin keeps these microtubules from growing too long by cutting these microtubules, and thus, effectively enhances turnover in spindles. In line with this model, our inference data (*Figure 8*) suggests the new plus ends of newly cut microtubules are mostly unstable. In line with this model, our inference scheme predicts values for $\alpha$ that are smaller than 0.5 for chromosome-proximal microtubules in all four metaphase spindles. However, given the small number of microtubules in close contact to chromosomes, some significant uncertainty on this number remains.

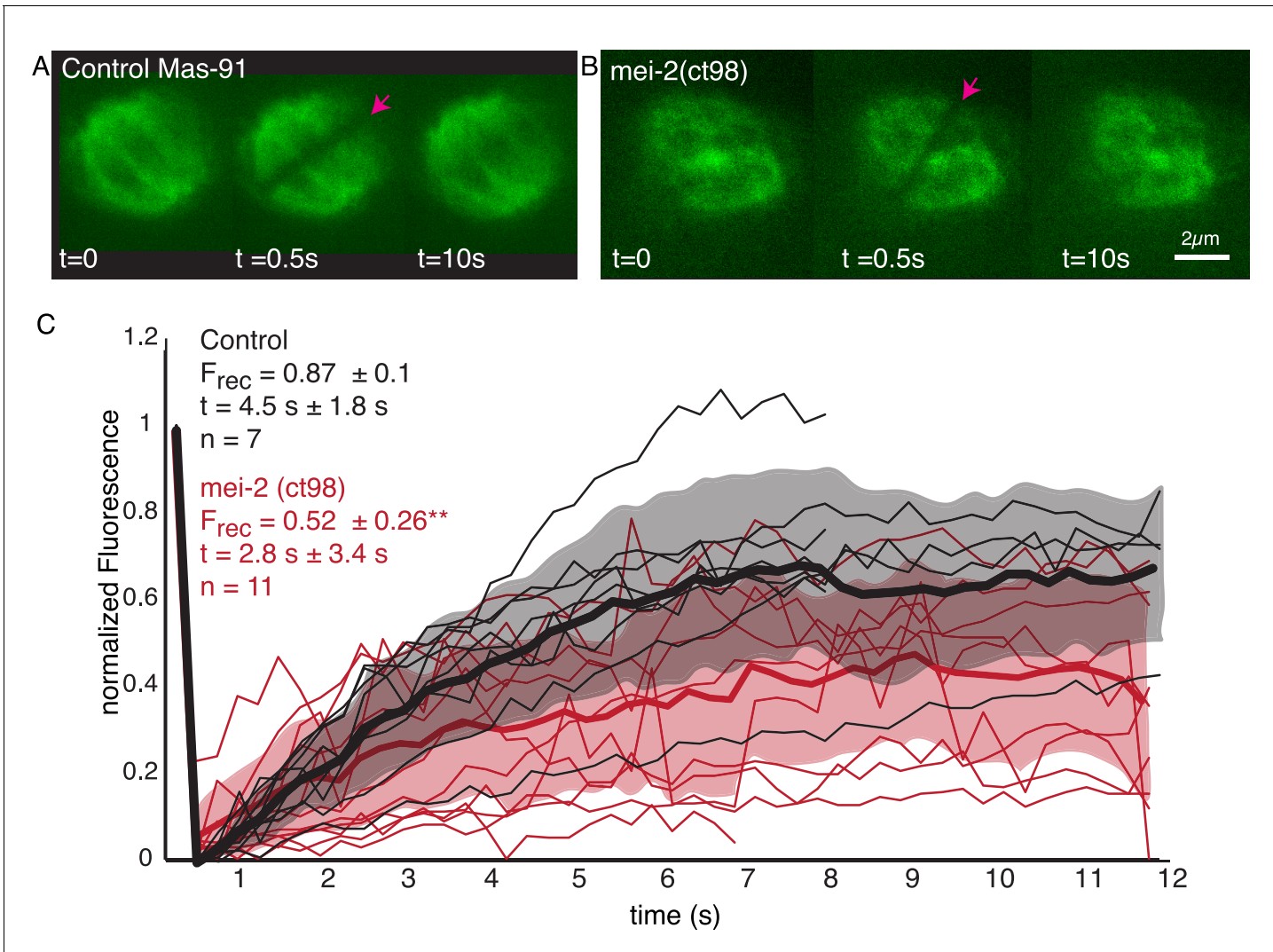

**Figure 9.** Microtubule turnover during metaphase of meiosis I in mei-2(ct98) embryos. (**A**) Light microscope images of spindles in control embryos in meiosis I prior to and after photobleaching at the spindle center (red arrow). The time points are indicated. (**B**) Light microscope images of spindles in meiosis I in the katanin mei-2(ct98) mutant prior to and after photobleaching at the spindle center (red arrow). The time points are indicated. Scale bar 2 μm. (**C**) Plot of the recovery of the bleach mark over time in control embryos (black) and mei-2 (ct98) (red) for different datasets. Mean values are indicated by thick lines, the shaded region corresponds to the standard deviation. The fraction of recovery ($F_{rec}$) and recovery times for the datasets are shown in the plot.

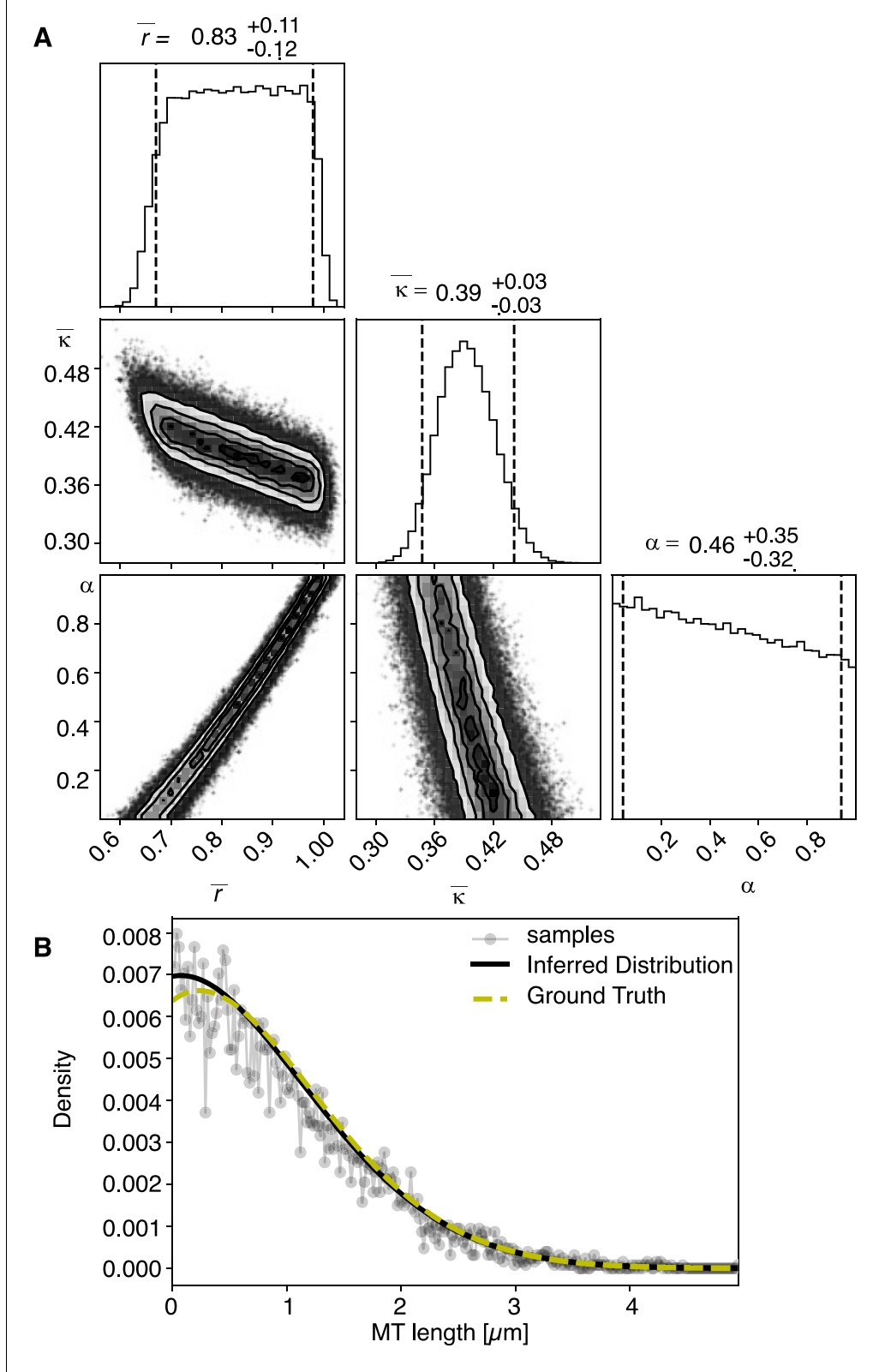

**Figure 10.** Results of inference versus ground truths on artificial data: case with cutting. (**A**) Likelihood distribution of model parameters determined by Markov Chain Monte Carlo (MCMC) for microtubules assuming cutting of microtubules. The top boxes show the totally marginalized distribution of parameters, with dashed lines delimiting the 95% confidence interval. Surface plots show cuts through the likelihood distributions, marginalized onto

*Figure 10 continued on next page*

*Figure 10 continued*

2D subspaces. Lines are contour lines, dots indicate MCMC samples. (B) Comparison of experimentally determined length distribution of microtubules (gray) to the prediction of the highest likelihood model (black) and the ground truth (green).

## Discussion

Acentrosomal meiotic spindles in *C. elegans* undergo a reorganization from a bipolar microtubule arrangement in metaphase to an inter-chromosomal microtubule array in anaphase. Along with this microtubule reorganization, chromosomes are segregated to extrude half of the genetic material as polar bodies. The underlying mechanism of the microtubule reorganization is not very well understood, and several mechanisms could be involved, for instance, katanin-mediated severing as reported for *C. elegans* meiosis I (*Joly et al., 2016*; *Srayko et al., 2006*), transport of microtubules as reported for *C. elegans* and *Xenopus* meiotic spindles (*Mullen and Wignall, 2017*; *Brugués et al., 2012*), or changes in microtubule polymerization dynamics a shown for *Xenopus* meiotic spindles (*Brugués et al., 2012*; *Needleman et al., 2010*). Here we have generated complete 3D reconstructions of meiotic spindles in *C. elegans* at different stages and combined this ultrastructural analysis with light microscopy and simulations to investigate the rearrangement of microtubules during meiosis.

Based on light microscopy, previous publications suggested that the microtubule rearrangement from metaphase to anaphase in meiosis I and II in *C. elegans* could be driven by an initial inward sliding of antiparallel microtubules by kinesin 14 family members (*McNally et al., 2016*) and a subsequent depolymerization of microtubules at the spindle pole due to severing by katanin (*McNally et al., 2006*). This was based on the observation that the initial phase of spindle shortening is accompanied by an increase of microtubule density, followed by a further shortening and depolymerization of microtubules at the spindle poles, resulting in a decrease of microtubule density. In agreement with this, our tomographic reconstructions show an initial increase in microtubule number and density during early anaphase, which is followed by a decrease in microtubule number and density in mid anaphase. However, while inward sliding of microtubules would result in an increased density, it does not explain the observed twofold increase of microtubule number. In addition, our data showed a poleward-directed movement of microtubules, contradicting an inward sliding. This suggests that inward sliding is unlikely to contribute to spindle shortening and cannot account for the appearance of microtubules between the chromosomes that characterizes anaphase.

The reorganization of microtubules could alternatively be driven by a selective depolymerization of microtubules at the spindle poles. However, our data suggests that the spindle rearrangement is mainly driven by changes in microtubule nucleation and turnover. A more local analysis of the changes in microtubule length and number, which revealed differences between microtubules near the chromosomes (within 150 nm) and those further away (*Figure 7*), as well as investigations of local changes using the mathematical model (*Figure 8*) showed spatial differences in microtubule properties and dynamics.

Our data showed that spindles are made from arrays of dynamic short microtubules, which turnover within 5 s. We further showed that the dramatic structural rearrangements observed from metaphase to anaphase are correlated with drastic changes in the microtubule number and length distribution. Meiotic metaphase spindles are composed of fewer but longer microtubules, while spindles in early anaphase are made of more but shorter microtubules.

These observations led us to ask whether severing of microtubules by katanin, or an increase in microtubule dynamics, allowing more nucleation and/or higher rates of catastrophe, would better explain our observations. For this, we developed a mathematical model that predicts the microtubule length distribution from cutting rates and turnover dynamics. We inferred parameters for the model using the microtubule length distribution and numbers found in electron tomography. This analysis severely constrains the possible mechanisms for spindle restructuring from metaphase to anaphase. Our data suggest that cutting of microtubules in the vicinity of chromosomes is important for maintaining metaphase spindles. It is tempting to speculate that microtubules near the chromosomes are more stable, similar to kinetochore microtubules in mitosis, and that katanin supports the turnover of those more stable microtubules by severing. For the transition from metaphase to anaphase, however, cutting seems to play less of a role. Globally the fraction of microtubules that show

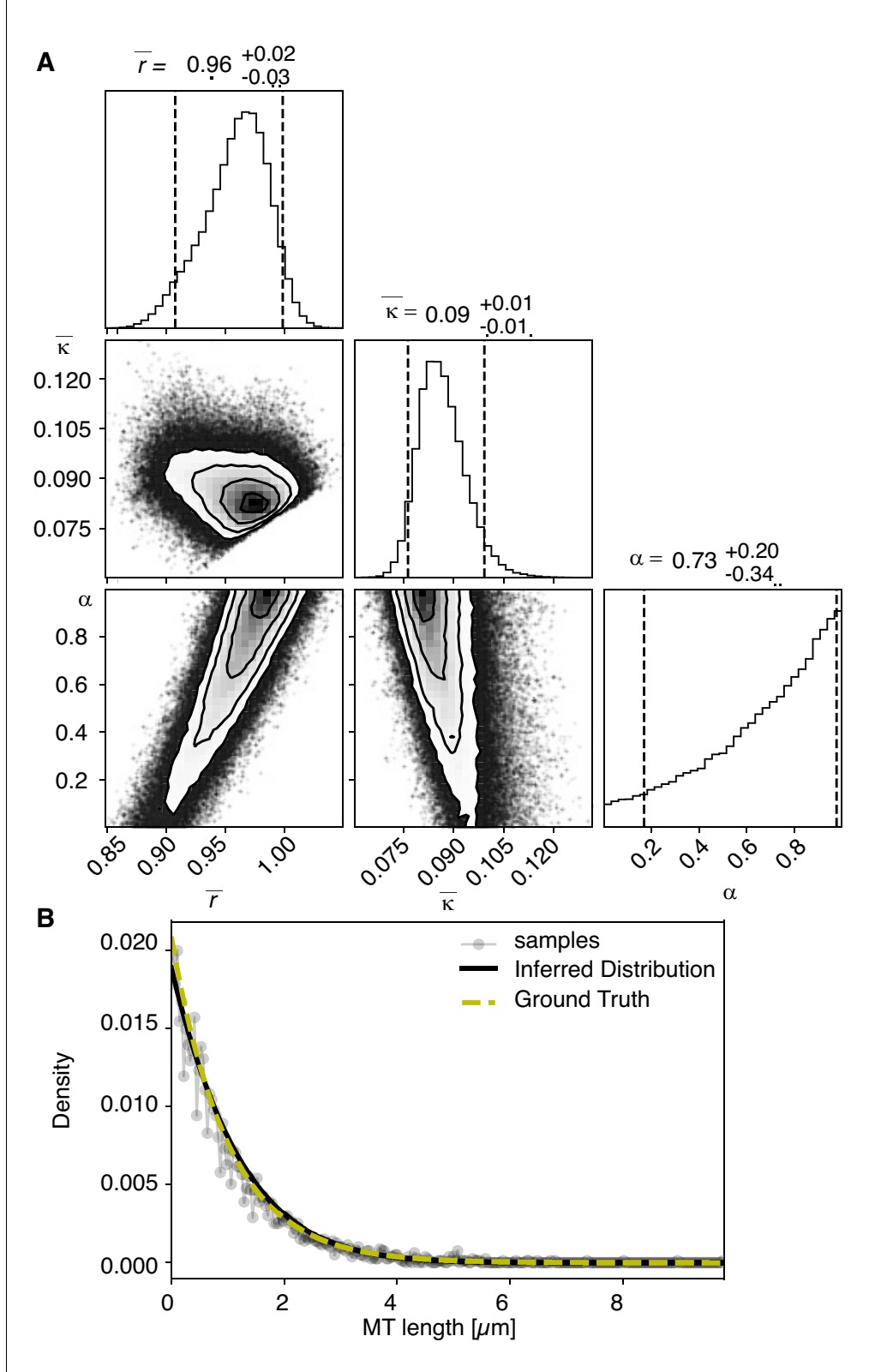

**Figure 11.** Results of inference versus ground truths on artificial data: case without cutting. (**A**) Likelihood distribution of model parameters determined by Markov Chain Monte Carlo (MCMC) for microtubules assuming microtubules are not cut. The top boxes show the totally marginalized distribution of parameters, with dashed lines delimiting the 95% confidence interval. Surface plots show cuts through the likelihood distributions, marginalized onto

*Figure 11 continued on next page*

*Figure 11 continued*

2D subspaces. Lines are contour lines, dots indicate MCMC samples. (**B**) Comparison of experimentally determined length distribution of microtubules (gray) to the prediction of the highest likelihood model (black) and the ground truth (green).

signs of cutting is small. Moreover, cutting seems to be much less prominent, even in chromosome-proximal microtubules during anaphase.

Interestingly, a role for microtubule-severing proteins, including katanin, in microtubule amplification has recently been suggested based on in vitro studies. Here, katanin induces nanoscale damage to the microtubule lattice, resulting in the incorporation of GTP tubulin (*Vemu et al., 2018*; *Schaedel et al., 2015*; *Schaedel et al., 2019*) and stabilization of microtubules. The incorporation of GTP tubulin into the lattice is thought to have two effects: promoting rescue as well as the stability of a new plus end formed upon severing. This would result in an amplification of microtubules. Consistent with this, our data shows a twofold increase of microtubule number during the transition from metaphase to early anaphase. Similarly, *Srayko et al., 2006* reported a decrease in microtubule number in *C. elegans* embryos depleted of katanin.

In summary, by combining light microscopy with electron tomography and mathematical modeling we analyzed the reorganization of microtubules during the transition from metaphase to anaphase in *C. elegans* meiotic embryos. Our data suggests that this reorganization is driven by changes in microtubule growth and/or turnover and that katanin promotes microtubule turnover by severing microtubules near the chromosomes.

# Materials and methods

## Key resources table

| Reagent type (species) or resource | Designation | Source or reference | Identifiers | Additional information |
|---|---|---|---|---|
| Strain, strain background (*Caenorhabditis elegans*) | MAS-91 | Martin Srayko (U of Alberta) | (unc-119(ed3) III; ltIs37 [pAA64; pie-1::mCherry::HIS58]; ruIs57[pie-1::GFP::tubulin+unc-119(+)]) | Expresses GFP tubulin and histone mCherry in the *C. elegans* embryo |
| Strain, strain background (*C. elegans*) | SA250 | CGC | (tjIs54 [pie-1p::GFP::tbb-2 +pie-1p::2xmCherry::tbg-1 +unc-119(+)]; tjIs57 [pie-1p::mCherry::his-48+unc-119(+)]) | Expresses GFP tubulin and histone mCherry in the *C. elegans* embryo |
| Strain, strain background (*C. elegans*) | FM13 | Frank McNally (UC Davis) | (*mei-2(ct98) I; ruIs57 [pAZ147: pie-1^promoter^:: tubulin::GFP; unc-119(+)]; itls37 [unc-119(+) pie-1^promoter^::mCherry::H2B]; him-8(e1489) IV*) | Mei-2(ct98) Katanin mutant |
| Software, algorithm | MATLAB, MathWorks | MATLAB R2020a, | | Including image processing toolbox |
| Software, algorithm | IMOD, Boulder Labs, CO | http://bio3d.colorado.edu/imod | | |
| Software, algorithm | Amira, FEI | AmiraZibEdition | | |

## Worm strains and gene silencing by RNAi

### Worm strains

The following *C. elegans* strains were used in this study: strain MAS91 (unc-119(ed3) III; ltIs37 [pAA64; pie-1::mCherry::HIS58]; ruIs57[pie-1::GFP::tubulin+unc-119(+)]) for live-cell imaging and correlative light microscopy/electron tomography, strain SA250 (tjIs54 [pie-1p::GFP::tbb-2+pie-1p::2xmCherry::tbg-1+unc-119(+)]; tjIs57 [pie-1p::mCherry::his-48+unc-119(+)]), MAS91 and FM13 (*mei-2(ct98) I; ruIs57 [pAZ147: pie-1^promoter^::tubulin::GFP; unc-119(+)]; itls37 [unc-119(+) pie-1^promoter^::mCherry::H2B]; him-8(e1489) IV*), for FRAP experiments. Strains were cultured and maintained at 16°C as described (*Brenner, 1974*).

## Light microscopy

### Sample preparation for light microscopy

Oocytes for live-cell imaging were prepared as described previously (*Woog et al., 2012*). For FRAP measurements, meiotic spindles in oocytes were observed in the uterus of adult hermaphrodites (strain SA250) mounted on a thin 4% Agarose pad between a slide and a coverslip. Polystyrene microspheres (Microspheres 0.10 µm, Polysciences, Inc) were added to the agar solution before specimen mounting to immobilize the living worms.

### Spinning disk confocal fluorescence imaging

Live imaging was performed using a spinning disk confocal microscope (Nikon Ti2000, Yokugawa CSU-X1), equipped with 488 nm and 561 nm diode lasers, an EMCCD camera (Hamamatsu), and a 60× water-immersion objective (CFI Plan Apo VC 60X WI, NA 1.2, Nikon). Acquisition parameters were controlled using a home-developed LabVIEW program (LabVIEW, National Instruments). Images were acquired every 1 s with a single z-plane.

### FRAP experiments

The photobleaching system was constructed on the above-mentioned spinning disk confocal microscope. 80 MHz femtosecond pulsed laser with 0.3 nJ pulse energy and 800 nm center wavelength was used for performing photobleaching and generated from a Ti:sapphire pulsed laser (Mai-Tai, Spectra-Physics, Mountain View, CA). The photobleaching laser was focused through the same objective for imaging, and photobleaching was performed by moving the sample on a piezo-stage (P-545 PIano XYZ, Physik Instrumente) in three dimensions controlled by a home-developed Lab-VIEW program (LabVIEW, National Instruments). Scanning line-bleaching with z-steps was created by moving the stage perpendicular to the pole-to-pole axis back and forth on the focal plane while lowering the stage in the z direction. The parameter for bleaching in length by depth was 6 × 2 µm. The moving speed of the stage was 50 µm/s.

### *mei-2(ct98)* FRAP experiment

Worms from the *mei-2(ct98ts)* strain as well as MAS-91 worms were kept at 16× until the time of imaging. The FRAP experiment was conducted using a Yokogawa CSU-W1 SoRa dual cam spinning disk confocal. This microscope is equipped with seven lasers for imaging (405, 445, 488, 514, 561, 594, 640 nm) that allow near-optimal excitation of common fluorescent proteins and minimize crosstalk. This spinning disk confocal is mounted on a Nikon Ti2 inverted microscope with a motorized stage for time-lapse imaging of multiple stage positions, piezo z-drive for rapid z-stack acquisition, and a Perfect Focus module to compensate for stage drift during time-lapse imaging. For image acquisition, the microscope is equipped with two types of cameras optimal for different types of experiments, which are mounted on the two camera ports of the CSU-W1 dual cam head: (1) a Hamamatsu ORCA Fusion camera for high-resolution imaging and (2) dual Photometrics Prime 95B cameras on a Cairn TwinCam for higher speed imaging and/or simultaneous acquisition of two channels. Hardware triggering allows rapid acquisition of z-stacks. The microscope is equipped with an Acal BFi UV Opti-Microscan point scanner that we used for the FRAP experiment. This system is integrated with Nikon NIS Elements software for seamless experimental setup and data acquisition. The movies were acquired with a 60 × 1.27 NA water objective and 2.8× SoRa magnifier with 100 ms exposure times and 250 ms intervals (4 frames/s).

### Analysis of fluorescence recovery after photobleaching and microtubule poleward flux

FRAP and rate of microtubule poleward flux were calculated with a combination of Fiji (*Schindelin et al., 2012*) and MATLAB (MATLAB and Statistics Toolbox Release 2012, The MathWorks, Nitick, USA). Time-lapse images of spindles expressing GFP::tubulin and mCherry::histone (corresponding to chromosomes) were realigned in a routine for matching, rotation, and translation using Rigid Body of Fiji's StackReg plug-in, so that the random displacement of the spindle due to the spontaneous motion of the worm was corrected.

Poleward flux and recovery of photobleached makers were tracked using a program written in MATLAB (MATLAB and Statistics Toolbox Release 2012, The MathWorks). Line scans of GFP-labeled tubulin along the metaphase spindle were extracted over the course of anaphase. To track the microtubule poleward flux, each line scan at each time point can be divided into two halves by the middle plane of the spindle. The half of the line profile with the bleached mark was subtracted from the other half of the non-bleached profile by a reflection of symmetry around the middle plane of the spindle. This profile subtraction was used to remove spatial variations in the background fluorescence, a valid procedure assuming mirror symmetry of the spindle around its middle plane. A Gaussian function was used to fit the subtracted profile to locate the center of the bleached mark, and thus the position of the bleached mark versus time was extracted. A straight line was fitted to the position of the bleached mark versus time to retrieve the rate of the bleached mark. The recovery time of GFP::tubulin after photobleaching was determined by using an exponential fit. The fluorescence intensities of photobleached marks were calculated by summing intensities over three pixels (~0.5 µm) around the center of the photobleach mark.

## Electron microscopy

### Sample preparation

Samples for electron tomography were prepared as described (*Woog et al., 2012*). Briefly, hermaphrodites were dissected in Minimal Edgar's Growth Medium (*Edgar, 1995*) and embryos in early meiosis were selected and transferred to cellulose capillary tubes (Leica Microsystems, Vienna, Austria) with an inner diameter of 200 µm. The embryos were observed with a stereomicroscope, transferred to membrane carriers at appropriate stages, and immediately cryo-immobilized using an EMPACT2 high-pressure freezer (Leica Microsystems) equipped with a rapid transfer system (*Pelletier et al., 2006*). Freeze substitution was performed over 3 days at −90℃ in anhydrous acetone containing 1% OsO$_4$ and 0.1% uranyl acetate using an automatic freeze substitution machine (EM AFS, Leica Microsystems). Epon/Araldite-infiltrated samples were then embedded in a thin layer of resin and polymerized for 3 days at 60℃. Embedded embryos were re-mounted on dummy blocks and serial semi-thick (300 nm) sections were cut using an Ultracut UCT Microtome (Leica Microsystems). Sections were collected on Formvar-coated copper slot grids and post-stained with 2% uranyl acetate in 70% methanol followed by Reynold's lead citrate.

### Electron tomography

For dual-axis electron tomography (*Mastronarde, 1997*), 15 nm colloidal gold particles (Sigma-Aldrich) were attached to both sides of semi-thick sections to serve as fiducial markers for subsequent image alignment. Series of tilted views were recorded using a TECNAI F30 transmission electron microscope (FEI Company, Eindhoven, The Netherlands) operated at 300 kV. Images were captured every 1.0° over a ± 60° range at a pixel size of 2.3 nm using a Gatan US1000 2K × 2K CCD camera. Using the IMOD software package, a montage of 2 × 1 (meiosis I: metaphase #1, metaphase #2, anaphase [late] #1, anaphase [late] #2; meiosis II: metaphase #1, metaphase #2) or 2 × 2 (meiosis I: anaphase [early]; meiosis II: anaphase [late]) frames was collected and combined for each serial section to cover the lengths of the meiotic spindles (*Kremer et al., 1996*; *Mastronarde, 1997*).

For image processing, the tilted views were aligned using the positions of the fiducials. Tomograms were computed for each tilt axis using the R-weighted back-projection algorithm (*Gilbert, 1972*). In order to cover the entire volume of each spindle, we acquired tomograms of about 8–12 consecutive sections per sample. In total, we recorded 10 wildtype spindles in meiosis I and II.

### Three-dimensional reconstruction and automatic segmentation of microtubules

We used the IMOD software package (http://bio3d.colorado.edu/imod) for the calculation of electron tomograms (*Kremer et al., 1996*). We applied the Amira software package for the segmentation and automatic tracing of microtubules (*Stalling et al., 2005*). For this, we used an extension to the filament editor of the Amira visualization and data analysis software (*Redemann et al., 2017*; *Redemann et al., 2014*; *Weber et al., 2012*). We also used the Amira software to stitch the obtained 3D models in z to create full volumes of the recorded spindles (*Redemann et al., 2017*;

*Weber et al., 2014*). The automatic segmentation of the spindle microtubules was followed by a visual inspection of the traced microtubules within the tomograms. Correction of the individual microtubule tracings included manual tracing of undetected microtubules, connection of microtubules from section to section, and deletions of tracing artifacts (e.g., membranes of vesicles). Approximately 5% of microtubules needed to be corrected (*Redemann et al., 2017*).

### Data and error analysis

Data analysis was performed using either the Amira software package or by exporting the geometric data of the traced microtubules followed by an analysis using MATLAB (MATLAB and Statistics Toolbox Release 2012, The MathWorks). In our analysis of spindle structure, the following errors were considered (*Redemann et al., 2017*). Briefly, during the data preparation and the imaging process, the tomograms are locally distorted. Furthermore, the exposure of the electron beam causes a shrinking of the sample. During the reconstruction of the microtubules, however, the most important errors occur in the tracing and matching process. In addition, the data is again distorted in all directions to align the tomograms. We assumed that this distortion primarily compensates the distortion of the imaging process. For the tracing, the error was previously analyzed for reconstructions of *C. elegans* centrosomes (*Weber et al., 2012*). We assumed that the error lies in the same range of 5–10%. In addition, the traced microtubules were manually verified. It is more difficult to estimate the error of the matching algorithm (*Weber et al., 2014*) since it depends on the local density and properties of the microtubules. The quality of our analysis should be influenced only by minor 3D distortions.

### Quantification of tomographic data

#### Microtubule length and positioning

The reconstruction algorithm for microtubules in serial section electron microscopy represents each microtubule as a 3D piecewise linear curve of its centerline. Thus, the length of the microtubule is given by the sum of the lengths of the line segments.

### Mathematical model and inference

#### Mathematical model for microtubule length distributions

The distribution $\psi(l)$ of microtubule lengths $l$ is set by the microtubule growth velocity $v_g$, their turnover rate $r$, and the rate at which microtubules are severed by proteins like katanin $\kappa$. It obeys

$$\partial_t \psi = -\partial_l (v_g \psi) - (r + \mathrm{K}l)\psi + \mathrm{K}(1+\alpha)\int_l^\infty dl'\, \psi\left(l'\right), \tag{1}$$

Note that the dimensions of $\kappa$ are per time per length, and thus the longer microtubules are more likely to get cut. The dimensionless parameter $\alpha$, which can take values from 0 to 1, determines the likelihood that a new microtubule plus end created by severing is stable. After some basic algebra, we find that the steady states of *Equation (1)* obey

$$v_g \partial_l^2 \psi + (\mathrm{r} + \mathrm{K}l)\partial_l \psi + (2+\alpha)\mathrm{K}\psi = 0 \tag{2}$$

with the boundary conditions

$$v_g \psi(0) = rN - \mathrm{K}\alpha N(l), \tag{3}$$

$$\partial_l v_g \psi|_{l=0} = -r\psi(0) + \mathrm{K}(1+\alpha)N, \tag{4}$$

which state that the creation of new microtubules by nucleation and severing balances the loss of microtubules by turnover such that the total number of microtubules *N* stays a constant.

Here, $(l) = (fl) = \frac{1}{N}\int_0^\infty dl'\, \psi'\left(l'\right)l^l$ denotes the average microtubule length.

For given parameters $r$, $\kappa$, $\alpha$, we can solve for the microtubule length $\psi(l)$ distribution numerically. Our custom written code uses second-order finite differences and is available from the authors upon reasonable request.

## Bayesian inference method for the relative importance of cutting and catastrophe

We seek to infer the probability distribution $P(model\,data)$ of the model parameters $(r, \alpha, \kappa)$, from the electron tomography data. This can be done using Bayes' formula

$$P(model|data) = P(data|model)P(model)/P(data), \qquad (5)$$

where $P(data|model)$ is the likelihood of the observed data given the model, $P(model)$ is the prior, and $P(data)$ is the finally marginalized likelihood. Since $P(data)$ is in general not easily calculated, we use Markov Chain Monte Carlo sampling – specifically the toolbox (*Foreman-Mackey, 2020*; https://github.com/dfm/emcee) – to approach this problem. This has the advantage that we need not provide an expression for $P(data)$. We next discuss the expressions that we use for $P(data|model)$ and $P(model)$, respectively.

### Calculating $P(data|model)$

This term calculates the likelihood that the measured data (the observed microtubule lengths in our case) are a draw from the probability distribution function predicted by the model. To calculate this term, we first solve *Equation 2* with the model parameters and obtain the probability distribution $p(l) = \psi(l)/\int_0^\infty dl\psi(l)$, which characterizes the length distribution in the model.

To obtain the likelihood of the data given the model, we sort the data in to 256 bins of equal size distributed from length 0 to 1.1 times the length of the longest microtubule seen in experiment. We then use $p(l)$ to determine the expectation value $\lambda_i$ for the number of microtubules found in bin $i$. (Note that we constrained the overall nucleation rate in the model to the value which would make the experimentally observed number of microtubules equal to the expected total number of microtubules.) With this, the probability of finding $m_i$ microtubules in bin $i$ is Poisson distributed, and thus

$$P(data|model) = \Sigma(i)\frac{\lambda_i^{m_i}e^{-\lambda_i}}{m_i!} \qquad (6)$$

Note that during this procedure we ignore data for microtubules that are shorter than 150 nm long since we suspect that many of these are artifacts from the tomography reconstruction method rather than real data points.

### Choosing a prior $P(model)$

We seek to not over-constrain our search space and thus use a so-called uninformative prior. This means that we assign all models within the parameter ranges $\bar{r}>0$, $\bar{\kappa}>0$ and $0<\alpha<1$ the same a priori likelihood.

### Test on generated test data

To test our inference scheme, we generated datasets from known distributions. We generated datasets that have comparable numbers and length of distributions as our experimental data and ran our inference scheme. In *Figures 10* and *11*, we show the results of this procedure and compare against ground truth for one case without and one case with cutting.

## Acknowledgements

The authors are thankful to Th. Müller-Reichert for continuous support and to S Tulok and Dr A Walther (Core Facility Cellular Imaging, Faculty of Medicine Carl Gustav Carus, TU Dresden) for help with light microscopy. The authors are also grateful to the members of the electron and light microscopy facility at the Max Planck Institute of Molecular Cell Biology and Genetics (MPI-CBG, Dresden) for technical assistance, J Baumgart (MPI-PKS, Dresden) for initial help in data analysis, and F McNally (UC Davis) for the *mei-2(ct98) C. elegans* strain. The authors would like to thank Drs E O'Toole and R McIntosh (Boulder) for a critical reading of the manuscript. IL and ES were supported by funds from the Deutsche Forschungsgemeinschaft (MU 1423/3-1, 3-2, and 8-1 to T M-R) and the Human Frontier Science Programme (RGP 0034/2010 to DN and TMR). SR received funding from the Faculty of Medicine Carl Gustav Carus of the TU Dresden (Frauenhabilitationsstipendium). The

work of CHY was supported by the National Science Foundation grant (DMR-0820484 and Neuro-Nex #1934288) and National Institutes of Health Grant (1R01GM104976-01).

## Additional information

### Funding

| Funder | Grant reference number | Author |
| --- | --- | --- |
| Deutsche Forschungsgemeinschaft | MU 1423/3-1 | Ina Lantzsch |
| Deutsche Forschungsgemeinschaft | MU 1423/3-2 | Ina Lantzsch |
| Deutsche Forschungsgemeinschaft | MU 1423/8-1 | Erik Szentgyoergyi |
| Technische Universität Darmstadt | Frauenhabilitation | Stefanie Redemann |
| Natural Sciences and Engineering Research Council of Canada | | Martin Srayko |
| National Science Foundation | DMR-0820484 | Che-Hang Yu |
| National Science Foundation | NeuroNex #1934288 | Che-Hang Yu |
| National Institutes of Health | 1R01GM104976-01 | Che-Hang Yu |
| Human Frontier Science Program | RGP 0034/201 | Yu-Zen Chen |

The funders had no role in study design, data collection and interpretation, or the decision to submit the work for publication.

### Author contributions

Ina Lantzsch, Data curation, Formal analysis, Writing - original draft; Che-Hang Yu, Conceptualization, Data curation, Formal analysis, Investigation; Yu-Zen Chen, Hossein Yazdkhasti, Data curation, Formal analysis; Vitaly Zimyanin, Data curation, Formal analysis, Writing - review and editing; Norbert Lindow, Steffen Prohaska, Resources, Software, Visualization; Erik Szentgyoergyi, Data curation; Ariel M Pani, Conceptualization, Resources, Data curation; Martin Srayko, Conceptualization, Resources, Supervision, Writing - original draft; Sebastian Fürthauer, Conceptualization, Data curation, Software, Formal analysis, Validation, Visualization, Writing - original draft, Project administration; Stefanie Redemann, Conceptualization, Data curation, Formal analysis, Supervision, Funding acquisition, Validation, Investigation, Visualization, Methodology, Writing - original draft

### Author ORCIDs

Che-Hang Yu ![ORCID] http://orcid.org/0000-0002-0353-9752
Stefanie Redemann ![ORCID] https://orcid.org/0000-0003-2334-7309

### Decision letter and Author response

Decision letter https://doi.org/10.7554/eLife.58903.sa1
Author response https://doi.org/10.7554/eLife.58903.sa2

## Additional files

### Supplementary files

• Transparent reporting form

## Data availability

Electron microscopy models of microtubules and chromosome surfaces will be made available on Dryad under https://doi.org/10.5061/dryad.x3ffbg7k5. Example data and analysis code is available at https://github.com/SebastianFuerthauer/SpindleRerrangement (copy archived at https://archive.softwareheritage.org/swh:1:rev:d558f2000a3186de466f6fa3491570298d4a3950).

The following dataset was generated:

| Author(s) | Year | Dataset title | Dataset URL | Database and Identifier |
|---|---|---|---|---|
| Redemann S | 2021 | *C. elegans* meiotic spindles | https://doi.org/10.5061/dryad.x3ffbg7k5 | Dryad Digital Repository, 10.5061/dryad.x3ffbg7k5 |

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
