## [Decision Letter]

**Acceptance summary:**

Using a combination of electron tomography, fluorescence microscopy and theoretical modelling, this study provides detailed information about microtubule numbers, lengths and turnover in the spindle of the nematode *C. elegans* at different stages of meiosis. Microtubules are short compared to spindle length, and their number peaks in early anaphase, although the total mass of polymerised tubulin decreases only later. This work provides new insight into the spindle structure at 'single microtubule resolution' and into microtubule dynamics during meiosis.

**Decision letter after peer review:**

Thank you for submitting your article "Microtubule re-organization during female meiosis in *C. elegans*" for consideration by *eLife*. Your article has been reviewed by 3 peer reviewers, one of whom is a member of our Board of Reviewing Editors, and the evaluation has been overseen by Anna Akhmanova as the Senior Editor. The following individual involved in review of your submission has agreed to reveal their identity: Erin Tranfield (Reviewer #2).

The reviewers have discussed the reviews with one another and the Reviewing Editor has drafted this decision to help you prepare a revised submission.

Summary:

In this manuscript, findings from tomographic datasets of 10 *C. elegans* meiotic spindles from metaphase and anaphase (early, mid, and late) spindles (6 MI and 4 MII) are presented. The focus of the manuscript is on the observation that the transition from metaphase to anaphase involves a significant reorganization of the structure in which the number of MTs increases and the mean length decreases 2-fold. The authors develop a mathematical model to assess the relative contributions of 1) changes in MT dynamics, and 2) increased MT severing activity to the reorganization phenomenon. The model explains the data by a global change in MT dynamics and, in fact, indicates that MT severing makes hardly any contribution to the MT shortening observed in anaphase. The work is timely and the topic is of great interest; the quality of the EM data is excellent and these data can be expected to become a valuable resources in the field.

Essential revisions:

1. To compare the model with the data, the authors "average away" a large amount of detailed information present in the EM data and make additional simplifying assumptions that may be questioned. For example, it may be an oversimplification to assume mono-modal length distributions in the model that can be described by averages. In figure 1B the metaphase spindles look like there are two populations. The situation in anaphase looks even more complicated particularly if there is a surge of nucleation at the start of anaphase generating new short MT. The detailed 3D data sets are simplified down to a single spatial dimension (the spindle axis) and single length estimator (the average). The authors should provide some evidence/do some tests to validate their approach. How sensitive are the predictions of the model to the simplifying assumptions made and to the averaging out of detail?

2. A major weakness of the manuscript is considered to be the lack of experimental test of the prediction of the model which the authors present as their main conclusion. It should be possible to perform FRAP experiments to test the effect of katanin mutants on microtubule turnover to confirm or contradict the main conclusion that the authors derive from their model and that in part argues against pervious work. There is a well-characterized (and fast acting) ts allele of MEI-1 called mei-1(or642) (O'Rourke et al., PLoS One 2011 and McNally et al., MBoC, 2014) that could be used to test the effect of katanin on microtubule turnover by FRAP.

3. The authors should please be a bit clearer which EM data sets are new and which ones were re-used from previous work (for example including this information in Table 1). The expectation would be that the information from the new datasets is also used in the theoretical analysis presented in this manuscript. The context to previous work by others could be explained more clearly by being more specific when presenting background in the introduction so that it will be easier to understand what's new and different here compared to previous work (particularly compared to Yu et al. 2019 and Srayko et al. 2006).

4. Technical concerns:

4.1. FRAP analysis: To which extent does flux versus microtubule polymerization/depolymerization contribute to recovery. Is using a mono-exponential function to fit the recovery curves justified given that flux may contribute to recovery? How are the FRAP data used in the model? Is the contribution from flux to recovery considered separately from the contribution of polymerization/ depolymerization?

4.2. p.10, 2nd paragraph: Is the observed decrease in average microtubule length really independent of position? What is the factor of decrease as a function of position? Is the notion of global vs local change really fully supported by the data?

4.3. Model: Are microtubule minus ends considered stable after severing? Alpha is introduced, but the authors do not seem to come back to it later. What is it?

4.4. Model: Throughout, it would be useful to provide confidence intervals for the values that the authors extract from their model or provide some other statistical measure for the reliability of the prediction.

4.5. Does the model make the same predictions for meiosis II spindles or is turnover regulation different there?

4.6. Model: On page 9, lines 15-17 – the authors claim that if all the dynamic parameters except nucleation do not change then the length distribution should not change. However, if there is a change in nucleation, there will be a short-term increase in short MT, thereby shifting the length distribution.

---

## [Author Response]

Essential revisions:1. To compare the model with the data, the authors "average away" a large amount of detailed information present in the EM data and make additional simplifying assumptions that may be questioned. For example, it may be an oversimplification to assume mono-modal length distributions in the model that can be described by averages. In figure 1B the metaphase spindles look like there are two populations. The situation in anaphase looks even more complicated particularly if there is a surge of nucleation at the start of anaphase generating new short MT. The detailed 3D data sets are simplified down to a single spatial dimension (the spindle axis) and single length estimator (the average). The authors should provide some evidence/do some tests to validate their approach. How sensitive are the predictions of the model to the simplifying assumptions made and to the averaging out of detail?

We apologize to the referees who have been misled (by us) into thinking that we only use averaged data to infer rates of cutting and turnover from the data. In this paper we used the measured lengths of all microtubules to infer the relative importance of cutting and microtubule catastrophe for setting the microtubule length distribution. The theory that we use to calculate the length distribution for given rates of cutting and catastrophe is generic. The Bayesian inference method (Markov Chain Monte Carlo Sampling) that we adapted to infer parameters of the theory uses the lengths of all microtubules, and not just the average length. We now better explain this procedure in an expanded methods section.

The referees are however correct in stating that we assumed that all microtubules obeyed the same growth rules (same cutting and catastrophe rates). In the original manuscript we had tested this assumption by including inferred rates separately for filaments whose center of masses where closer to the poles and closer to the chromosomes, and showing that these where not statistically different. We now further subdivided the data, in particular we separated out all microtubules that approach the chromosomes closer than 150nm. This turns out to be a very small subpopulation of microtubules in metaphase, less than 25% of microtubules. Interestingly this small subpopulation has a markedly different length distribution than the other MTs, which shows clear signatures of cutting. We thank the referees for motivating us to push our analysis further in this regard.

2. A major weakness of the manuscript is considered to be the lack of experimental test of the prediction of the model which the authors present as their main conclusion. It should be possible to perform FRAP experiments to test the effect of katanin mutants on microtubule turnover to confirm or contradict the main conclusion that the authors derive from their model and that in part argues against pervious work. There is a well-characterized (and fast acting) ts allele of MEI-1 called mei-1(or642) (O'Rourke et al., PLoS One 2011 and McNally et al., MBoC, 2014) that could be used to test the effect of katanin on microtubule turnover by FRAP.

We thank the referees for this suggestion. We have conducted FRAP experiments in the FM13 mei-2(ct98) strain, which has a reduced microtubule severing rate, and included those results in the paper (Figure 9). This experiment revealed that katanin does promote the turn-over of microtubules in meiotic *C. elegans* spindles, possibly by acting on microtubules near the chromosomes (150nm). Together with our new data on the different length distribution of microtubules within 150nm from the chromosomes this additional experiment has helped to significantly improve this manuscript.

3. The authors should please be a bit clearer which EM data sets are new and which ones were re-used from previous work (for example including this information in Table 1). The expectation would be that the information from the new datasets is also used in the theoretical analysis presented in this manuscript. The context to previous work by others could be explained more clearly by being more specific when presenting background in the introduction so that it will be easier to understand what's new and different here compared to previous work (particularly compared to Yu et al. 2019 and Srayko et al. 2006).

We thank the referees for this suggestion and have included this information in Table 1 and Table 2. We have also combined the data from meiosis I and II in the main figures to clarify that the information of all available datasets was used for this manuscript. In addition to this we updated the relevant text passages to better explain previous analysis that were done on some of the datasets.

4. Technical concerns:4.1. FRAP analysis: To which extent does flux versus microtubule polymerization/depolymerization contribute to recovery. Is using a mono-exponential function to fit the recovery curves justified given that flux may contribute to recovery? How are the FRAP data used in the model? Is the contribution from flux to recovery considered separately from the contribution of polymerization/ depolymerization?

We thank the reviewers for this comment. In order to distinguish possible effects of microtubule flux from recovery due to microtubule polymerization/depolymerization we have re-analyzed our FRAP data using larger regions for the analysis. For this we analyzed the recovery rate in a 2µm wide box. Within the recovery time of approximately 10s a microtubule could slide 1µm (based on the velocity of the measured poleward flux). This distance of 1µm is much smaller than the box size, arguing that the recovery is mainly caused by microtubule polymerization/depolymerization.

The FRAP results complement the inference scheme, but are not used in it. Our Bayesian inference scheme infers the relative importance of microtubule cutting and catastrophe for the overall length distribution, i.e. the ratio of two time scales. It does so using only the measured lengths of microtubules. In contrast FRAP measures the absolute time scale of microtubule turnover. This additional information can be used to estimate absolute turnover time scales. We clarify this in the model description.

4.2. p.10, 2nd paragraph: Is the observed decrease in average microtubule length really independent of position? What is the factor of decrease as a function of position? Is the notion of global vs local change really fully supported by the data?

We thank the reviewers for raising this point. We have re-analyzed our data and updated the figure 3 accordingly. As the spindles change significantly in length from metaphase to late anaphase we have normalized the spindle length across meiosis in these new plots (Figure 3) to better determine the decrease as a function of position. We still find that the length of microtubules changes everywhere along the spindle axis, however microtubule in the center of the spindles show a bigger change in length than microtubules at the poles.

4.3. Model: Are microtubule minus ends considered stable after severing? Alpha is introduced, but the authors do not seem to come back to it later. What is it?

We included alpha in the theory and sought to infer it from length data. For the global population, given that cutting turns out to be rare, the value of alpha cannot be substantially constrained from the data. This is shown in Figure 6, Figure 6- Supplementary Figure 1 In essence, all values of alpha are equally consistent with the global data.

However, in the revised manuscript we also separately analyzed the MTs contacting the chromosomes (closer than 150nm) and find that these show clear signs of cutting. For these microtubules we infer that alpha is likely closer to 0 than to 1 – meaning that newly created minus ends are most likely unstable, see Figure 8 and Figure 8 – Supplementary Figure 1 and 2 We now discuss this in the appropriate section of the paper.

4.4. Model: Throughout, it would be useful to provide confidence intervals for the values that the authors extract from their model or provide some other statistical measure for the reliability of the prediction.

We give the errors as standard error to the mean in the text. The figures also include values for the 5% and 95% quantile of the distribution of all model parameters. Finally, the density plots in Figure 6 and 8 and Figure 6 – Supplementary Figure 1, Figure 8 – Supplementary Figure 1 and 2 show the confidence surfaces of parameters in their three dimensional phase space spanned by rbar, kappabar, alpha. We now take more time to explain these plots. We also emphasize that this way of reporting error goes beyond the usual standards of our field.

4.5. Does the model make the same predictions for meiosis II spindles or is turnover regulation different there?

We now include the analysis and data for meiosis two. In short, all our results hold for meiosis I and II alike.

4.6. Model: On page 9, lines 15-17 – the authors claim that if all the dynamic parameters except nucleation do not change then the length distribution should not change. However, if there is a change in nucleation, there will be a short-term increase in short MT, thereby shifting the length distribution.

The referee is right that a sudden change in the nucleation rate will in the short term change the length distribution. This change would subsist for time scales that are comparable to MT turnover times – i.e. seconds. In contrast the reconstruction of the filament network progresses slowly – over several minutes. We thus operate under the assumption that whatever we measure is the reflection of a quasi-steady state. In other words, we assume that the rate of change of parameters is slow compared to a microtubule lifetime.